# Memorization and the Orders of Loss: A Learning Dynamics Perspective

## Abstract

Deep learning has become the de facto approach in nearly all learning tasks. It has been observed that deep models tend to memorize and sometimes overfit data, which can lead to compromises in performance, privacy, and other critical metrics. In this paper, we explore the theoretical foundations that connect memorization to various orders of sample loss, i.e., sample loss, sample loss gradient, and sample loss curvature, focusing on learning dynamics to understand what and how these models memorize. To this end, we introduce two proxies for memorization: Cumulative Sample Loss (CSL) and Cumulative Sample Gradient (CSG). CSL represents the accumulated loss of a sample throughout training, while CSG is the gradient with respect to the input, aggregated over the training process. CSL and CSG exhibit remarkable similarity to stability-based memorization, as evidenced by considerably high cosine similarity scores. We delve into the theory behind these results, demonstrating that CSL and CSG represent the bounds for stability-based memorization and learning time. Additionally, we extend this framework to include sample loss curvature and connect the three orders, namely, sample loss, sample loss gradient, and sample loss curvature, to learning time and memorization. The proposed proxy, CSL, is four orders of magnitude less computationally expensive than the stability-based method and can be obtained with zero additional overhead during training. We demonstrate the practical utility of the proposed proxies in identifying mislabeled samples and detecting duplicates where our metric achieves state-of-the-art performance. Thus, this paper provides a new tool for analyzing data as it scales in size, making it an important resource in practical applications.

## 1 Introduction

Deep learning has become the de facto standard for almost all machine learning tasks from image (Ho et al., 2020) and text generation (Radford et al., 2019) to classification (Krizhevsky et al., 2009; Soufleri et al., 2024a) and reinforcement learning (Shakya et al., 2023). While they have been extremely successful, they tend to memorize and overfit to the training data. While some memorization is indeed needed to obtain generalization (Feldman, 2020), these deep models can also memorize totally random images (Zhang et al., 2017). Thus to understand memorization, researchers have put in significant effort (Zhang et al., 2017; Arpit et al., 2017; Carlini et al., 2019a; Feldman & Vondrak, 2019; Feldman & Zhang, 2020; Feldman, 2020). Such focus is crucial due to the broad implications of memorization for multiple connected areas, including generalization (Zhang et al., 2021; Brown et al., 2021), noisy learning (Liu et al., 2020), identifying mislabeled examples (Maini et al., 2022), recognizing rare and challenging instances (Carlini et al., 2019a), ensuring privacy (Feldman, 2020), and addressing risks from membership inference attacks (Shokri et al., 2017; Carlini et al., 2022; Ravikumar et al., 2024b).

Many approaches to study memorization have been proposed (Carlini et al., 2019a; Jiang et al., 2021; Feldman, 2020). Notably, the stability-based metric proposed by Feldman (2020) measures the change in expected output probability when the sample under investigation is removed from the training dataset. This metric offers a robust theoretical framework for understanding memorization, which was subsequently validated empirically for deep neural networks (Feldman & Zhang, 2020). However, this approach is impractical for most applications due to its high computational cost. Recent literature has introduced other proxies for memorization, such as learning time (Jiang et al.,

Figure 1: Solid lines represent our contributions in this paper, linking various orders of loss–namely, loss, loss gradient with respect to input, and curvature of loss with respect to input–to memorization and learning time. The dashed line represents the previously established static link.

2021), adversarial distance (Del Grosso et al., 2022), model confidence (Carlini et al., 2019b), and input loss curvature (Garg et al., 2024; Ravikumar et al., 2024a). While these proxies have been successful in understanding the memorization behavior of neural networks, most fail to capture certain properties of memorization such as bi-modality (Lukasik et al., 2023). Thus, establishing a strong theoretical foundation of memorization and its proxies is of critical importance.

While prior work has investigated the properties of the loss function, such as input curvature post-training and its connection to memorization, we establish a theoretical framework that explains how learning dynamics drive the similarity between the orders of loss (loss, loss gradient w.r.t input and loss curvature w.r.t input), memorization and learning time. We propose two new proxies for memorization: Cumulative Sample Loss (CSL) and Cumulative Sample Gradient (CSG) to capture information from training dynamics. CSL represents the total loss of a sample accumulated over the entire training process, while CSG is the gradient of the loss with respect to the input, aggregated throughout training. The proposed CSL proxy is *4 orders of magnitude* less computationally expensive than stability-based (Feldman & Zhang, 2020) memorization and $\approx 14\times$ less expensive than input loss curvature (Garg et al., 2024). It is important to note that the *14×* estimate is conservative, as CSL can be obtained for free during training, making the computational benefits even greater than these numbers suggest.

We validate our theory with experiments and show that the proposed cumulative metrics have very high cosine similarity with the memorization score from Feldman & Zhang (2020). Further, we show that the proposed metrics can be used to identify duplicates and mislabeled examples; notably, the adaptation of our proposed metrics leads to achieving state-of-the-art performance in these applications. In summary, our contributions are:

- We present a new theoretical framework that links learning dynamics, memorization, and the three orders of loss (loss, gradient of loss w.r.t input and curvature of loss w.r.t input) as shown in Figure 1. Specifically, we establish novel connections between sample loss, sample gradient, and sample curvature, relating them to learning time and memorization.

- We propose two new memorization proxies: Cumulative Sample Loss (CSL) and Cumulative Sample Gradient (CSG). These proxies demonstrate high similarity to stability-based memorization methods but are significantly more computationally efficient, offering a reduction in computational cost by several orders of magnitude.

- We validate our theory through experiments on deep vision models, demonstrating the efficacy of CSL and CSG as memorization proxies.

- We showcase the practical applications of our metrics in identifying mislabeled examples and duplicates in datasets, achieving state-of-the-art performance in these tasks.

## 2 NOTATION AND BACKGROUND

**Notation.** We denote random variables using bold capital letters $\mathbf{V}$, their instances as italic small letters $v$ for scalars, $\vec{v}$ for vectors, and capital letters $V$ for matrices. For simplicity and compactness, we ignore the notation when vectors are in the subscript, for example $\nabla_w = \nabla_{\vec{w}}$. Consider a learning problem, where the task is learning the mapping $f : \vec{x} \mapsto y$ where $\vec{x} \sim \mathbf{X} \in \mathbb{R}^n$ and $y \sim \mathbf{Y} \in \mathbb{R}$. A dataset $S = (\vec{z_1}, \vec{z_2}, \ldots, \vec{z_m}) \sim \mathbf{Z}^m$ consists of $m$ samples, where each sample $\vec{z_i} = (\vec{x_i}, y_i) \sim \mathbf{Z}$.

We also use a leave one out set which the the dataset $S$ with the $i^{th}$ sample removed denoted by $S^{\setminus i} = (\vec{z_1}, \ldots, \vec{z_{i-1}}, \vec{z_{i+1}}, \ldots, \vec{z_m})$. We use $g_S^\phi \sim \mathbf{G}_S$ to denote the function learnt by the neural network by the application of a possibly random training algorithm $\mathcal{A}$, on the dataset $S$ where $\phi \sim \mathbf{\Phi}$ denotes the randomness of the algorithm. Let $\vec{w}_t^k$ denote the weights of the $k^{th}$ layer at iteration $t$. Consider a single data sample $\vec{x_i} = [x_{i1} \quad x_{i2} \quad \cdots \quad x_{in}]^T$ represented as a column vector. Then a dataset or mini-batch with $m$ examples is represented as $X = [\vec{x_1} \quad \vec{x_2} \quad \cdots \quad \vec{x_m}]$. A cost function $c : \mathbf{Y} \times \mathbf{Y} \to \mathbb{R}^+$ is used to evaluate the performance of the model. The cost at a sample $\vec{z_i}$ is referred to as the loss $\ell$ evaluated at $\vec{z_i}$, defined as $\ell(g, \vec{z_i}) = c(g(\vec{x_i}), y_i)$. Typically, we are interested in the loss of $g$ over the entire data distribution, called the population risk, which is defined as $R(g) = \mathbb{E}_z[\ell(g, \vec{z})]$. Since the data distribution $\mathbf{Z}$ is generally unknown, we instead evaluate the empirical risk as follows $R_{\text{emp}}(g, S) = \frac{1}{m} \sum_{i=1}^m \ell(g, \vec{z_i}), \vec{z_i} \in S$.

**Error Stability** of a randomized algorithm $\mathcal{A}$ for some $\beta > 0$ is defined as in Kearns & Ron (1997):

$$\forall i \in \{1, \cdots, m\}, \ \left| \mathbb{E}_{\phi, z}\left[\ell(g_S^\phi, z)\right] - \mathbb{E}_{\phi, z}\left[\ell(g_{S^{\setminus i}}^\phi, \vec{z})\right] \right| \leq \beta, \tag{1}$$

**Memorization** of the $i^{th}$ element $\vec{z_i} = (\vec{x_i}, y_i)$ in the dataset $S$ by an algorithm $\mathcal{A}$ is as:

$$\text{mem}(S, \vec{z_i}) = \left| \Pr_\phi[g_S^\phi(\vec{x_i}) = y_i] - \Pr_\phi[g_{S^{\setminus i}}^\phi(\vec{x_i}) = y_i] \right| \tag{2}$$

where the probability is taken over the randomness of the algorithm $\mathcal{A}$. We adapt the formulation from Feldman (2020) to ensure that the score remains positive, aligning with the practical method used for score calculation.

**Input Loss Curvature.** Following the curvature notation from prior works (Moosavi-Dezfooli et al., 2019; Ravikumar et al., 2024a; Garg et al., 2024), input loss curvature is defined as the sum of the eigenvalues of the Hessian $H$ of the loss with respect to input $\vec{z_i}$. This can be expressed using the trace as $\text{Curv}_\phi(\vec{z_i}, S) = \text{tr}(H) = \text{tr}(\nabla_{z_i}^2 \ell(g_S^\phi, \vec{z_i}))$.

**L-Bounded Loss.** We say that loss a loss function is L-bounded if it satisfies $0 \leq \ell \leq L$.

$\alpha$**-adjacency.** A dataset $S$ is said to contain $\alpha$-adjacent elements if it contains two elements $z_i, z_j$ such that $z_j = z_i + \epsilon$ for some $\epsilon \in B_p(\alpha)$ (read as $\alpha$-Ball). Note that this can be ensured through construction. Consider a dataset $S'$ which has no $z_j$ s.t $z_j = z_i + \epsilon; z_j, z_i \in S'$. Then we can construct $S$ such that $S = \{z \mid z \in S'\} \cup \{z_i + \epsilon\}$ for some $z_i \in S', \epsilon \in B_p(\epsilon)$, ensuring $\alpha$-adjacency holds. See additional discussion in Section 4.2.3 on its validity for real applications.

$\lambda$**-Proximal.** Let $\ell(\vec{w}_0)$ represent the initial training loss. Then, there exists a $\lambda$-Proximal iteration $T_p$ if $\ell(\vec{w}_{T_p}) = (1 - \lambda)\ell(\vec{w}_0)$ for some $\lambda$. In our theoretical framework, we assume that the optimizer used is Stochastic Gradient Descent (SGD). For conciseness, background information on SGD, as well as additional background on Lipschitz continuity, uniform model bias, generalization and bounded gradients, are provided in Appendix A.

## 3 RELATED WORK

Memorization in deep neural networks has gained attention, with recent works improving our understanding of its mechanisms and implications (Zhang et al., 2017; Arpit et al., 2017; Carlini et al., 2019a; Feldman & Vondrak, 2019; Feldman, 2020; Feldman & Zhang, 2020; Maini et al., 2022; Lukasik et al., 2023; Garg et al., 2024; Ravikumar et al., 2024a). This research is driven by the need to understand generalization (Zhang et al., 2017; Brown et al., 2021; Zhang et al., 2021), identify mislabeled examples (Pleiss et al., 2020; Maini et al., 2022), and detect out-of-distribution or rare sub-populations (Carlini et al., 2019a; Ravikumar et al., 2023). Additionally, memorization impacts privacy (Dwork et al., 2006; Feldman, 2020; Soufleri et al., 2024b), robustness (Shokri et al., 2017; Carlini et al., 2022), and unlearning (Kurmanji et al., 2023; Kodge et al., 2024).

Previous studies explored learning dynamics from different angles. Mangalam & Prabhu (2019) showed that deep networks first learn simple samples, Pruthi et al. (2020) analyzed the influence of training examples, and Toneva et al. (2019) studied forgetting during training. Maini et al. (2022) introduced split learning and forgetting times, while Carlini et al. (2019a) combined metrics to study memorization. Jiang et al. (2021) proposed the C-score, a computationally efficient memorization

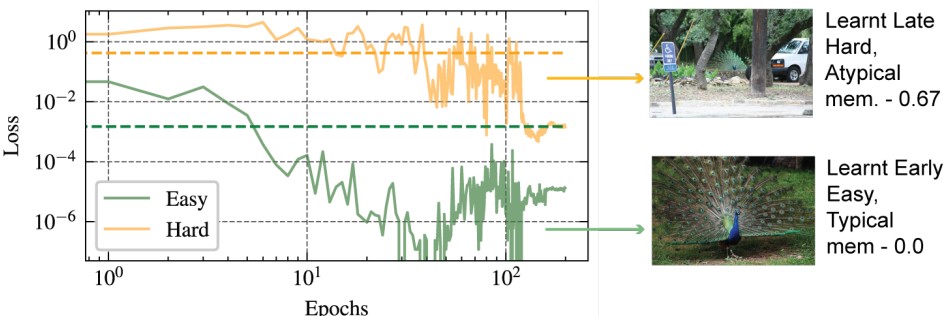

Figure 2: Learning Dynamics: Figure depicts how learning time affects average loss. Average loss is visualized as dashed line and the loss values are visualized in solid plot. Easy examples are typical less memorized, hard atypical examples are memorized more.

proxy. More recently, Garg et al. (2024) used input loss curvature as a proxy for stability-based memorization scores (Feldman, 2020), supported by theoretical analysis in Ravikumar et al. (2024a), though both focused on post-training analysis. In contrast, this paper investigates input loss curvature, sample loss, and sample loss gradients over training. Thus, providing a broader perspective on the dynamics of learning and its relation to learning time and memorization.

## 4 LEARNING DYNAMICS AND MEMORIZATION

### 4.1 PROPOSED MEMORIZATION PROXIES

To build intuition, let us explore the loss progression of two samples, namely, an "easy" and a "hard" example, both from the same class (peacock) in the ImageNet dataset (Russakovsky et al., 2015), as illustrated in Figure 2. In this context, loss refers to the per-sample cross-entropy loss, which tracks how well the model predicts a specific example at each stage of training.

For the easy sample, learned early in the process, the loss follows a simple pattern: it starts high, quickly drops, and stays low for the rest of the training. The hard sample, on the other hand, behaves differently. Its loss remains high for a much longer period before eventually dropping, indicating that it is learned much later. This contrast in loss dynamics' patterns evidently shows how the cumulative per-sample loss throughout training can distinguish between easy and hard examples with precision.

Additionally, the figure demonstrates traditional metrics like learning time and forgetting time, which rely on thresholds to determine when a sample is learned, may fall short of distinguishing between easy and hard examples. As we observe in Figure 2, a sample might be learned in one epoch, unlearned in the next, and then relearned showing that learning is a noisy process. To overcome this noise, we propose using cumulative sample loss, CSL (or mean sample loss) and cumulative sample gradient, CSG (or mean sample gradient) as more reliable metrics. These metrics smooth out the noise from fluctuations in learning. As we will demonstrate, hard examples tend to be memorized by the model, while easy examples are generalized; the proposed cumulative metrics, tracked throughout the training process, are key to capturing this correlation. The two proposed metrics CSL and CSG of a sample $\vec{z}_i$ can be formally defined as:

$$\text{CSL}(\vec{z}_i) = \sum_{t=0}^{T_{max}} \ell(\vec{w}_t, \vec{z}_i), \quad \text{CSG}(\vec{z}_i) = \sum_{t=0}^{T_{max}} \|\nabla_{\vec{z}_i} \ell(\vec{w}_t)\|_2^2 \quad (3)$$

where $T_{max}$ is the total number of iterations of SGD.

### 4.2 THEORETICAL ANALYSIS

To formalize this intuition, we introduce the concept of the sample learning condition. In optimization theory, the necessary condition for optimality for an unconstrained problem is typically expressed as $\nabla \ell(w) = 0$. In the case of optimizers like gradient descent or its extensions, convergence is typically characterized in terms of gradient norm given as $\|\nabla_w \ell(w_t)\|_2 \leq \tau$, where $\ell$ is the function to minimize and $\tau$ denotes an arbitrarily small threshold. Thus, as a natural extension of

this perspective, we define the sample learning condition as:

$$\frac{1}{T}\sum_{t=0}^{T-1}\|\nabla_{z_i}\ell(\vec{w}_t)\|_2^2 \leq \tau, \tag{4}$$

where the gradient is with respect to the sample $\vec{z}_i$. We interpret Equation 4 as follows: A sample is considered learned if the average sample loss gradient over the course of training falls below a certain threshold. Formally a sample is considered learned if the average per iteration gradient norm falls below a threshold $\tau$. As we will demonstrate, this formulation simplifies the ensuing mathematical expressions and analyses. In our pursuit of formalizing the key role of learning dynamics in memorization, we first examine the relationship between learning time and cumulative loss. We use the Stochastic Gradient Descent (SGD) optimizer for our analysis (see Appendix A for background on SGD). As an intermediate step, it is necessary to first analyze the convergence of the input gradient, which is discussed in Appendix B as Theorem B.1 due to space constraints in the main body of the paper. For convenience we group a set of assumptions below.

*SGD Convergence Assumptions.* The convergence of SGD in gradient norm holds under the $\mathcal{L}$-Lipschitz continuity of the loss function (Eq. 16), with a bounded gradient norm (Eq. 18) and an unbiased gradient estimator. Next, we present one of our key theoretical results, formally relating memorization and sample learning time.

### 4.2.1 LEARNING TIME LINKS TO CSL AND CSG

**Theorem 4.1** (Memorization upper bounds Learning Time). *Under the assumptions of SGD convergence, $\beta$-stability and L-bounded loss, if stochastic gradient descent (SGD) is performed for $T_{max}$ iterations, with the expected learning time for a reference sample denoted by $\hat{T}_{ref}$ and the loss estimation variance given by $\sigma_l$, then with confidence at least $1 - \delta$, the expected learning time for a sample $\vec{z}_i$ is bounded by its memorization score.*

$$\mathbb{E}_\phi[T_a] - \hat{T}_{ref} \leq \frac{T_{max}L}{\mathbb{E}_\phi[\ell(\vec{w}_0)]}\left(\text{mem}(a) + \frac{2\beta}{L} + \frac{2\sigma_l}{L\sqrt{\delta}}\right) \tag{5}$$

**Sketch of Proof.** This proof connects memorization with learning time by analyzing input gradient convergence using the sample learning condition. It compares the learning times of two samples, showing that their learning times are proportional to the difference in loss by using the convergence upper bound established in Theorem B.1. We leverage the result from Ravikumar et al. (2024a) to link sample's loss difference to memorization. The proof is available in Appendix E.2.

**Interpreting Theory.** In the theorem, $\hat{T}_{ref}$ refers to a reference sample that can be chosen so its learning time is nearly zero. To build intuition we can approximate $\mathbb{E}_\phi[\ell(\vec{w}_0)] \approx L$ and $\sigma_l \approx 0$. Based on these assumptions, the expected learning time is a fraction of the total number of iterations, $T_{max}$, with this fraction determined by the sample's memorization score. Thus we can interpret from the theorem that samples with higher memorization scores take longer to learn, meaning their expected learning time is directly linked to how much they are memorized.

**Theorem 4.2** (Cumulative loss bounds learning time). *Let the assumptions for SGD convergence hold, and let $T_{max}$ denote the maximum number of iterations of SGD. Further, assume there exists a $\lambda$-proximal reference sample and loss estimation variance is $\sigma_l$. Then, with a confidence $1 - \delta$, the learning time $T_{z_i}$ for any sample $\vec{z}_i \in S$ follows:*

$$T_{z_i} \leq T_{ref}\frac{\text{CSL}(\vec{z}_i)}{\lambda\ell(\vec{w}_0) + \ell(\vec{w}^*)\,T_{ref} - \sigma_l\,\delta^{-0.5}\,T_{ref}} \tag{6}$$

**Sketch of Proof.** We leverage an intermediate result from Theorem 4.1, which establishes a relationship between the learning times of two samples and their respective losses. The central step involves telescoping the loss differences, demonstrating that both the total loss decrease and the cumulative loss are proportional to the learning time of the sample. The proof is provided in Appendix E.3.

**Interpreting Theory.** The theorem shows that a sample's learning time is tied to its CSL, which tracks the total loss accumulated over time. The learning time is upper-bounded by a fraction of the reference sample learning time. This theorem captures the intuition that harder samples, with higher cumulative losses, take longer to learn, while easier samples, with lower cumulative losses, are learned faster.

**Theorem 4.3** (Cumulative sample gradient bounds learning time). *Under the SGD convergence assumptions, let $T_{max}$ denote the maximum number iterations. Assume the loss function satisfies the $\mu$-PL condition and there exists $\lambda$-proximal reference sample. Define $\mathcal{M}$ as the product of the second transformation constant $k_{gw}$ (Result 36) and $\mu$. Under these conditions, the learning time of a sample $\vec{z}_i$ is bounded by the cumulative sum of the input gradients throughout the training process.*

$$T_{z_i} \leq \frac{\mathcal{M} T_{max}}{2\lambda \ell(\vec{w_0})} \text{CSG}(\vec{z}_i) \tag{7}$$

**Sketch of Proof.** The proof leverages Theorem 4.2 and the PL condition to bound learning time using the gradient norm. The transformation result (see result 36) is applied to convert the weight gradient norm to the sample gradient norm. The full proof is provided in Appendix E.4.

**Interpreting Theory.** This theorem establishes a linear relationship between learning time and the proposed metric, CSG, demonstrating that the upper bound on learning time is linearly related to the input gradient. The upper bound is a fraction of the total iterations, $T_{max}$, where the fraction is determined by the ratio of CSG to the initial loss, scaled by the input-weight gradient transformation constant, the PL constant $\mu$, and the loss bound parameter on a reference sample.

### 4.2.2 MEMORIZATION LINKS

**Theorem 4.4** (Memorization bounds Cumulative Loss). *Assume the loss function is L-bounded, and the assumptions for the convergence of SGD hold. Additionally, let the error stability condition (Eq. 1) be satisfied. Then, with confidence $1 - \delta$, the memorization $\text{mem}(S, \vec{z}_i)$ of any sample $\vec{z}_i \in S$ satisfies the following inequality:*

$$C_5 \left( \text{CSL}(\vec{z}_i) - C_6 + C_7 \right) \leq \text{mem}(S, \vec{z}_a), \quad C_5 = \left( \frac{3L^2}{\mathbb{E}_\phi \left[ \ell(\vec{w_0}) \right]} - L \right)^{-1} \tag{8}$$

$$C_6 = \hat{T}_{ref} \ell(\vec{w}^{\,*}), \quad C_7 = \left( \beta - \frac{3L^2}{\mathbb{E}_\phi \left[ \ell(\vec{w_0}) \right]} - 3L - \left( 1 + \frac{6L}{\mathbb{E}_\phi \left[ \ell(\vec{w_0}) \right]} \right) \frac{\sigma_l}{\sqrt{\delta}} \right) \tag{9}$$

**Sketch of proof.** We start with Equation 33, which relates the learning time and loss difference of two samples, $\vec{z}_i$ and $\vec{z}_b = \vec{z}_{ref}$. Using the stability assumption and bound on the gradients, the proof derives lower bounds on the loss over multiple iterations. These bounds incorporate memorization terms from Theorem 4.1 and use Chebyshev's inequality to account for variance in loss estimation. Finally, the proof concludes by showing that memorization of sample can be bounded using constants that depend on the properties of the loss function and stability. The full proof is available in Appendix E.7.

**Discussion.** This theorem provides an upper bound on CSL by utilizing memorization, showing that CSL has a linear relationship with memorization. To build intuition around this result, we can interpret $C_6$ as representing a lower bound on the cumulative loss at each step, meaning the term $\text{CSL} - C_6$ measures how far the cumulative loss at each step is from this lower bound. $C_7$ accounts for the total estimation error, while $C_5$ is a scaling factor that adjusts the result to the appropriate range.

**Theorem 4.5** (Input Gradient bounds Memorization). *Let the assumptions of error stability 1, generalization 14, and uniform model bias 15 hold and assume the that the loss is L-bounded and satisfies $\mathcal{L}$-Lipschitz. Further assume the dataset is $\alpha$-adjacent. Then with probability at least $1 - \delta$*

$$\text{mem}(S, \vec{z}_i) \leq C_3 + C_4 \, \mathbb{E}_\phi \left[ \| \nabla \ell(g_{S \setminus i}^\phi, z_i) \| \right] \tag{10}$$

$$C_3 = \frac{m\beta}{L} + \frac{(4m-1)\gamma}{L} + \frac{2(m-1)\Delta}{L} + \frac{\mathcal{L}}{2L} \|\alpha\|^2, \quad C_4 = \frac{\|\alpha\|}{L} \tag{11}$$

**Sketch of Proof.** The proof uses stability and generalization assumptions, showing that the difference in loss when a sample is removed from the training set is influenced by the loss gradient. By applying the Lipschitz continuity of the loss function, the proof concludes that memorization is proportional to the gradient norm. Thus, the larger the gradient at the end of training for a sample, the more it is likely memorized during training. Full proof available in Appendix E.6.

**Interpreting Theory.** This theorem establishes a connection between memorization and the sample gradient at the end of training, demonstrating that the sample gradient serves as an upper bound on sample memorization. The theorem predicts a linear relationship between memorization and the sample gradient. The constants in this linear relationship include the stability term $\beta$, the model bias term $\Delta$, the generalization term $\gamma$, as well as the Lipschitz constant and the sample ball parameter $\alpha$ (i.e., there exist two samples within an $\alpha$-ball of each other).

### 4.2.3 LEARNING TIME, CSG AND SAMPLE CURVATURE

**Theorem 4.6** (Input Curvature bounds Input Gradient which bounds Learning Time). *Assume that the convergence assumptions for SGD hold, and that the loss function satisfies the $\mu$-PL condition. Additionally, assume that the Hessian of the loss is $\rho$-Lipschitz continuous, and the gradient variance is bounded by $\sigma^2$. Furthermore, assume the dataset contains a $\lambda$-proximal sample. Under these conditions, the learning time for a sample $\vec{z}_i$ is limited by the gradient of the input, which is itself bounded by the input curvature as follows*

$$T_{z_i} \leq \frac{\mathcal{M}T_{max}}{2\lambda\ell(\vec{w_0})} \sum_{t=0}^{T_{z_i}-1} \|\nabla_X \ell(w_t)\|_2^2 \leq C_1 + C_2 \sum_{t=0}^{T_{z_i}-1} \|\nabla_X^2 \ell(w_t)\| \tag{12}$$

$$C_1 = \frac{\eta^2 k_g \Gamma^3 \rho \mathcal{M}T_{max}^2}{12\lambda\ell(\vec{w_0})} + \frac{\mathcal{M}T_{max}k_g}{2\lambda\eta}, \quad C_2 = \frac{\eta k_g k_h (\sigma^2 + \Gamma^2)\mathcal{M}T_{max}}{2\lambda\ell(\vec{w_0})} \tag{13}$$

**Sketch of Proof.** The proof leverages the Lipschitz continuity of the Hessian to bound the change in loss during gradient descent, involving both gradient and curvature terms. Summing over iterations shows that the cumulative gradient is bounded by the cumulative curvature, thus bounding the total learning time. The full proof in available in Appendix E.5.

**Interpreting Theory.** This theorem establishes a link between learning time, CSG, and cumulative sample curvature. It shows that learning time is bounded by CSG, which in turn is bounded by cumulative sample curvature. The theorem implies that CSG provides a tighter bound on learning time than sample loss curvature. This is supported by the mislabel detection performance of CSG compared to cumulative sample curvature, as demonstrated in the Experiment Section 5.3.

**Remark on Assumptions.** We briefly and qualitatively evaluate the practicality of our assumptions. Prior work (Hardt et al., 2016) has demonstrated that models trained using stochastic gradient methods, such as stochastic gradient descent, exhibit low generalization error. Furthermore, it has been established that these methods are uniformly stable (Hardt et al., 2016), supporting the plausibility of our assumptions on stability (Equation 1) and generalization (Equation 14). Model bias is an intrinsic characteristic of the model itself, and it is reasonable to assume a uniform bound across different datasets. Virmaux & Scaman (2018) have provided a general upper bound for the Lipschitz constant of any differentiable deep learning model, validating the Lipschitz continuity assumption in the context of deep models. The assumptions of an unbiased gradient estimator, along with bounded gradient norm and variance, are widely used in the optimization literature (Lian et al., 2017; Aketi et al., 2024), making these assumptions reasonable. In practice, loss functions are often upper-bounded, supporting the validity of the bounded loss assumption. Finally, $\alpha$-adjacent dataset can be guaranteed by design. However, in practice, this may not be necessary since the size of the ball $B_p(\alpha)$ is unrestricted. Therefore, two samples from the same class that are 'similar' might suffice to meet this criterion.

**Key Theory Takeaways.** (1) Learning time exhibits a linear relationship with all three metrics (i.e., orders of loss). (2) Stability-based memorization also follows this linear relationship with the three metrics. (3) Additionally, learning time and memorization are linearly related. (4) Loss serves as the most compute-efficient proxy among the proxies considered for measuring memorization.

## 5 EXPERIMENTS

### 5.1 VALIDATING THEORY

In this section, we conduct experiments to empirically validate the theoretical relationships established in the paper. Specifically, we investigate the following connections: (1) the relationship

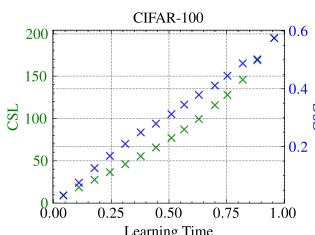 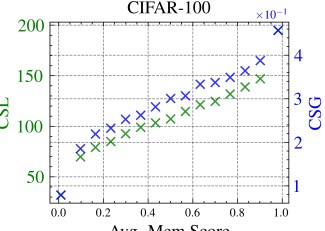 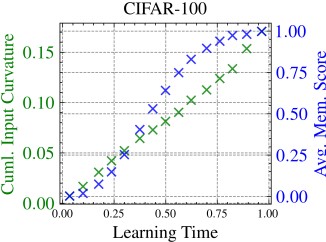

Figure 3: Learning time vs CSL and CSG on CIFAR-100 dataset.

Figure 4: Memorization score vs CSL and CSG on CIFAR-100 dataset.

Figure 5: Learning time vs memorization and input loss curvature on CIFAR-100.

between learning time and the three metrics—CSL (Cumulative Sample Loss), CSG (Cumulative Sample Gradient), and Cumulative Input Loss Curvature, and (2) the relationship between memorization, CSL, CSG, and learning time.

**Experiment.** We train a ResNet18 model (He et al., 2016) on the CIFAR-100 dataset (Krizhevsky et al., 2009), calculating learning time, CSL, CSG, and cumulative input loss curvature for each sample in the training set. For memorization scores, we utilize the precomputed stability-based memorization scores from Feldman & Zhang (2020). We plot a binned scatter plot of these metrics (see Appendix C.4 for more details on the setup).

**Results.** The results are visualized in Figures 8, 9 and 10. Figure 8 plots the CSL and CSG vs learning time. Figure 9 plots the the memorization score for CIFAR-100 from Feldman & Zhang (2020) vs CSL and CSG. And finally Figure 10 plots learning time vs memorization and cumulative input loss curvature.

**Takeaways.** Our theoretical results, particularly Theorems 4.1 - 4.6, predict a linear relationship between learning time, memorization, and the three orders of loss—loss, loss gradient, and loss curvature. The experimental results, as visualized in Figures 8, 9 and 10 empirically confirm these linear relationships. The slight non-linearity observed between learning time and memorization is likely due to the assumption of bounded loss in the theoretical framework, whereas the cross-entropy loss used in practice does not have a uniform bound across all subpopulations (see Ravikumar et al. (2024a) for a similar discussion).

## 5.2 SIMILARITY WITH MEMORIZATION

**Experiment.** In this section, we examine how well our proposed proxies–CSL, CSG, and loss curvature–correlate with the memorization score defined by Feldman & Zhang (2020). We conduct this experiment by training ResNet18 models on both CIFAR-100 and ImageNet datasets. For each dataset, we compute the memorization proxies and measure their cosine similarity with the memorization scores publicly made available by Feldman & Zhang (2020). This setup mirrors the approach used in Garg et al. (2024). Please see Appendix C.1 for additional setup details.

**Results.** The results are presented in Table 1, which shows the cosine similarity between the proxies and the Top-K memorized examples, as well as all examples. "Top 5K" refers to selecting the 5000 most memorized examples based on Feldman & Zhang (2020), and the cosine similarity for these examples is reported. The table compares three metrics: CSL, CSG, and curvature (Garg et al., 2024). For CIFAR-100, CSL emerges as the best proxy, while for ImageNet, it ranks a close second. Interestingly, for ImageNet, the most memorized examples exhibit a stronger correlation with the CSG. Across all samples on ImageNet, curvature has a very slight advantage over CSL.

**Takeaways.** CSL serves as the best proxy for CIFAR-100 and effectively captures memorization for both ImageNet's top-K examples and the entire dataset. Additionally, CSL proves to be highly computationally efficient, as it is available without extra computation during training. In comparison, it is approximately $14\times$ faster than curvature and 4 orders of magnitude faster than stability-based memorization, making it an attractive option for practical use (see Appendix C.3 for detailed compute cost breakdown). Additional results on various architectures (see Appendix C.2) show that these results are consistent across different network architectures.

| Dataset | Samples | Cosine Sim. w/ Mem. | | |
| | | Metric | | |
| | | CSL | CSG | $\nabla^2 \ell$ |
|---|---|---|---|---|
| CIFAR-100 | Top 5K | **0.94** | 0.86 | 0.90 |
| | All | **0.88** | 0.77 | 0.82 |
| ImageNet | Top 50K | 0.92 | **0.94** | 0.87 |
| | All | 0.71 | 0.68 | **0.72** |

Table 1: Cosine similarity match with FZ scores for CIFAR-100 and ImageNet.

Figure 6: 32 highest scores for CSL (left) and CSG (right) on clean CIFAR-100 reveal conflicting labels, such as Baby and Girl or Crab and Spider outlined in red.

| Dataset | Method | 1% Noise | 2% Noise | 5% Noise | 10% Noise |
|---|---|---|---|---|---|
| CIFAR-10 | Learning Time (LT) | $0.4951 \pm 0.0248$ | $0.4954 \pm 0.0044$ | $0.4911 \pm 0.0071$ | $0.4948 \pm 0.0057$ |
| | In Conf. (Carlini et al., 2019a) | $0.8781 \pm 0.0177$ | $0.8072 \pm 0.0130$ | $0.7254 \pm 0.0214$ | $0.6528 \pm 0.0042$ |
| | CL (Northcutt et al., 2021) | $0.8651 \pm 0.0127$ | $0.8905 \pm 0.0115$ | $0.8874 \pm 0.0019$ | $0.8551 \pm 0.0030$ |
| | SSFT (Maini et al., 2022) | $0.9626 \pm 0.0018$ | $0.9551 \pm 0.0020$ | $0.9498 \pm 0.0042$ | $0.9360 \pm 0.0020$ |
| | Curv. (Garg et al., 2024) | $0.9715 \pm 0.0045$ | $0.9776 \pm 0.0033$ | $0.9800 \pm 0.0003$ | $0.9819 \pm 0.0006$ |
| | CSL (Ours) | $\mathbf{0.9845 \pm 0.0026}$ | $\mathbf{0.9864 \pm 0.0004}$ | $\mathbf{0.9870 \pm 0.0003}$ | $\mathbf{0.9869 \pm 0.0005}$ |
| | CSLT (Ours) | $0.9501 \pm 0.0427$ | $0.9528 \pm 0.0401$ | $0.9433 \pm 0.0509$ | $0.9274 \pm 0.0689$ |
| | CSG (Ours) | $0.9681 \pm 0.0054$ | $0.9754 \pm 0.0029$ | $0.9783 \pm 0.0009$ | $0.9809 \pm 0.0011$ |
| CIFAR-100 | Learning Time (LT) | $0.5256 \pm 0.0012$ | $0.5227 \pm 0.0100$ | $0.5161 \pm 0.0051$ | $0.5203 \pm 0.0029$ |
| | In Conf. (Carlini et al., 2019a) | $0.7258 \pm 0.0102$ | $0.7236 \pm 0.0047$ | $0.7069 \pm 0.0069$ | $0.6884 \pm 0.0053$ |
| | CL (Northcutt et al., 2021) | $0.8723 \pm 0.0208$ | $0.8838 \pm 0.0006$ | $0.8733 \pm 0.0010$ | $0.8536 \pm 0.0006$ |
| | SSFT (Maini et al., 2022) | $0.8915 \pm 0.0045$ | $0.8893 \pm 0.0013$ | $0.8784 \pm 0.0030$ | $0.8664 \pm 0.0024$ |
| | Curv. (Garg et al., 2024) | $0.9856 \pm 0.0009$ | $0.9865 \pm 0.0011$ | $0.9876 \pm 0.0021$ | $0.9892 \pm 0.0012$ |
| | CSL (Ours) | $\mathbf{0.9891 \pm 0.0003}$ | $\mathbf{0.9895 \pm 0.0002}$ | $0.9895 \pm 0.0001$ | $0.9897 \pm 0.0001$ |
| | CSLT (Ours) | $0.9846 \pm 0.0059$ | $0.9857 \pm 0.0049$ | $0.9860 \pm 0.0045$ | $0.9865 \pm 0.0041$ |
| | CSG (Ours) | $0.9880 \pm 0.0007$ | $0.9888 \pm 0.0004$ | $\mathbf{0.9896 \pm 0.0008}$ | $\mathbf{0.9904 \pm 0.0006}$ |

Table 2: Evaluating the performance of mislabeled detection of the proposed framework against existing methods on CIFAR-10 and CIFAR-100 datasets under various levels of label noise.

## 5.3 MISLABELED DETECTION

**Experiment.** In this section, we leverage insights from our theoretical framework to develop a practical method for detecting mislabeled examples in training datasets. We evaluate the effectiveness of our approach by comparing it to several state-of-the-art methods for label error detection. The experiments are conducted on CIFAR-10 and CIFAR-100, where varying levels of symmetric label noise are introduced. Specifically, labels are randomly flipped to another class, uniformly across all classes, excluding the true label. To assess performance, we employ the Area Under the Receiver Operating Characteristic (AUROC) metric, which measures the ability of each method to correctly identify mislabeled examples under different noise conditions. Additional details of the experiments and the baseline techniques are available in Appendix C.1 and C.5.

**Results.** The results are presented in Table 2, showcasing the performance of each method on CIFAR-10 and CIFAR-100 at symmetric label noise levels of 1%, 2%, 5%, and 10%. The term LT (Learning Time) refers to the first epoch at which a sample is correctly classified (Jiang et al., 2021; Maini et al., 2022), reflecting the epoch at which the model learns a particular sample. This is contrasted with CSLT (Cumulative Sample Learning Time), which represents the cumulative count of epochs during which the sample is correctly predicted, essentially tracking the learning dynamics over time. The results in Table 2 clearly indicate that incorporating learning dynamics significantly improves performance in detecting mislabeled examples.

**Takeaways.** On CIFAR-10, the proposed CSL proxy consistently outperforms other methods in detecting mislabeled examples. However, on CIFAR-100, as the noise level increases, the gradient proxy, CSG gradually surpasses CSL in performance. This shift can be explained by the increased complexity in identifying mislabeled examples at higher noise levels, where higher-order information, such as gradients, becomes crucial for accurately detecting label errors.

**Compute Cost.** Compared to other techniques, CSL and CSG incur zero additional computational overhead. In contrast, Confidence Learning denoted as CL (Northcutt et al., 2021) requires training multiple (k-fold) models, with 3-folds used in this case, significantly increasing the computational cost. SSFT (Maini et al., 2022) requires training at least two subsets of the original training set,

Low CSG     High CSG

Figure 7: Using the proposed metric (CSG) uncovers a bias in the FMNIST dataset: darker clothing with lower contrast is often identified as high CSG (i.e. harder).

| Method | CIFAR-10 | CIFAR-100 |
|---|---|---|
| LT | $0.7029 \pm 0.0058$ | $0.7419 \pm 0.0059$ |
| In Conf. | $0.9237 \pm 0.0114$ | $0.8623 \pm 0.0131$ |
| CL | $0.5533 \pm 0.0031$ | $0.5873 \pm 0.0090$ |
| SSFT | $0.8490 \pm 0.0034$ | $0.7938 \pm 0.0045$ |
| Curv. | $0.9536 \pm 0.0030$ | $0.9639 \pm 0.0030$ |
| CSL (Ours) | $\mathbf{0.9821 \pm 0.0006}$ | $\mathbf{0.9886 \pm 0.0008}$ |
| CSG (Ours) | $0.9496 \pm 0.0022$ | $0.9715 \pm 0.0028$ |
| CSLT (Ours) | $0.9680 \pm 0.0034$ | $0.9870 \pm 0.0005$ |

Table 3: Result of duplicate detection using the proposed methods and other baselines on CIFAR-10 and CIFAR-100 datasets.

approximately doubling the training cost relative to standard training. Input loss curvature is computationally expensive, requiring about 14 times more compute than CSL. Methods such as LT (Learning Time), CSLT (Cumulative Sample Learning Time), CSG and CSL are roughly equivalent in terms of computational cost (CSG has a larger memory footprint). Therefore, CSL emerges as the most efficient method, providing the best balance between compute cost and performance.

## 5.4 DUPLICATE DETECTION

**Experiment.** In this section, we apply the proposed memorization proxies to identify duplicate examples in the dataset. We conduct two types of analysis: first, a qualitative analysis of duplicate detection on the unmodified CIFAR-100 dataset; second, a quantitative experiment where we intentionally introduce duplicates (250 duplicates) into the dataset and use our proxies to identify them. We use a ResNet18 (He et al., 2016) model for this experiment, and the performance of our method is evaluated against other techniques using the AUROC metric (see Appendix C.1 for setup details).

**Results.** The qualitative analysis results are presented in Figure 6, which demonstrates the detection of duplicates in the clean CIFAR-100 dataset. The quantitative experimental results are provided in Table 3, where we report the AUROC scores for our method compared to other techniques.

**Takeaways.** As shown in Figure 6, both CSL and CSG effectively identify the majority of duplicates in the unmodified CIFAR-100 dataset. This is further validated in Table 3, where we evaluate the performance of the method after intentionally introducing duplicates. Here, we observe that CSL consistently achieves the best performance in identifying duplicates across both the CIFAR-10 and CIFAR-100 datasets.

**Note on Hard Examples.** Additionally, the proposed metrics demonstrate the capability to identify unintended biases within the dataset. This is illustrated in Figure 7, where the CSG metric is applied to the FashionMNIST (Xiao et al., 2017) dataset. The figure visualizes images with low and high CSG values, revealing that images with high CSG scores, which are learned later during training, tend to be darker and have lower contrast. The poor performance or increased difficulty in learning such images may not accurately reflect real-world distributions. *To improve reproducibility, we have provided the code for all the experiments in the supplementary material.*

## 6 CONCLUSION

This paper provides a comprehensive theoretical framework that connects memorization proxies, such as input loss curvature, cumulative sample loss, and cumulative sample gradient, to learning dynamics and stability-based memorization. Our results demonstrate that these proxies are not only highly effective in capturing memorization behavior but also computationally efficient, being four orders of magnitude faster than existing stability-based metrics. We validate our framework through extensive experiments and show its practical applications in identifying mislabeled examples, bias, and duplicates in datasets. The proposed metrics achieve state-of-the-art performance in identifying duplicate and mislabeled examples. By offering efficient tools to understand memorization, our framework can lead to more interpretable models across a wide range of machine learning tasks. Ultimately, this work paves the way for more scalable, accurate, and data-centric approaches in the deployment of deep learning models in real-world applications.

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

APPENDIX

# A  ADDITIONAL BACKGROUND

In this section we provide a brief description of useful concepts used in the paper.

**Generalization.** An algorithm $\mathcal{A}$ generalizes with confidence $\delta$ and rate $\gamma'(m)$ if:

$$\Pr[|R_{emp}(g, S) - R(g)| \leq \gamma'(m)] \geq \delta. \tag{14}$$

**Uniform Model Bias.** The hypothesis $g_S^\phi$ learnt from applying algorithm $\mathcal{A}$ to learn the true conditional $f = \mathbb{E}[y|\vec{x}]$ is said to have a uniform bound on model bias given by $\Delta$ if:

$$\forall S \sim \mathbf{Z}^m, \quad \left|\mathbb{E}_\phi[R(g_S^\phi) - R(f)]\right| \leq \Delta. \tag{15}$$

**$\mathcal{L}$-Lipschitz Gradient.** The gradient of the loss function $\ell$ is said to be $\mathcal{L}$-Lipschitz continuous on $\mathbf{Z}$ if, for all $\vec{z_1}, \vec{z_2} \in \mathbf{Z}$, there exists a constant $\mathcal{L} > 0$ such that:

$$\|\nabla_{z_1}\ell(\vec{z_1}) - \nabla_{z_2}\ell(\vec{z_2})\| \leq \mathcal{L}\|\vec{z_1} - \vec{z_2}\| \tag{16}$$

**$\rho$-Lipschitz Hessian.** The Hessian of $\ell$ is said to be $\rho$-Lipschitz continuous on $\mathbf{Z}$ if, for all $\vec{z_1}, \vec{z_2} \in \mathbf{Z}$ and for all $S, g \in \mathbf{Z}^m, \mathbf{G}_S$ where $\mathbf{G}_S = \text{Range}(\mathcal{A}(S))$, there exists some $\rho > 0$ such that:

$$\|\nabla_{z_1}^2\ell(g, \vec{z_1}) - \nabla_{z_2}^2\ell(g, \vec{z_2})\| \leq \rho\|\vec{z_1} - \vec{z_2}\|. \tag{17}$$

**Bounded Gradient.** Suppose that for each iteration $t$, the expected cubic norm of the gradient with respect to the parameters $\vec{w_t}^k$ is bounded by a constant $\Gamma^3$, i.e.,

$$\mathbb{E}_t\left[\|\widetilde{\nabla}_{w_t^k}\ell(\vec{w_t}^k)\|_2^3\right] \leq \Gamma^3, \tag{18}$$

**$\mu$-PL (Polyak-Łojasiewicz) Condition.** Consider a function $f$, which is smooth and needs to be minimized without constraints. Assume there exists at least one minimizer $\vec{w}^*$ (not necessarily unique). Even if $f$ is not convex, it satisfies the Polyak-Łojasiewicz (PL) condition if there exists a constant $mu > 0$ such that:

$$\frac{1}{2}\|\nabla_w\ell(\vec{w})\|_2^2 \geq \mu\left[\ell(\vec{w}) - \ell(\vec{w}_*)\right] \tag{19}$$

**Loss Estimation Variance $\sigma_l$.** In practical applications, empirical loss is often approximated using a finite set of samples. Specifically, we estimate the expected loss as follows:

$$\mathbb{E}_z[\ell(\vec{w}, z)] \approx \frac{1}{M}\sum_{i=1}^{M}\ell(\vec{w}, \vec{z_i}) \tag{20}$$

The variance of this approximation from the true expected loss is referred to as the loss estimation variance, denoted by $\sigma_l$.

**Stochastic Gradient Descent (SGD).** In stochastic gradient descent (SGD), the model parameters (or weights) $\vec{w_t}$ at iteration $t$ are updated based on the gradient of the loss function $\ell(\vec{w_t}, \vec{z_i})$ with respect to the parameters, evaluated using a mini-batch or a single random sample $\vec{z_i}$ from the dataset. The general update rule for SGD is given by:

$$\vec{w}_{t+1} = \vec{w}_t - \eta_t\tilde{\nabla}_{\vec{w}}\ell(\vec{w}_t, \vec{z_i}) \tag{21}$$

Where $\eta_t$ is the learning rate at iteration $t$, $\tilde{\nabla}_{\vec{w}}\ell(\vec{w}_t, \vec{z_i})$ is the unbiased stochastic gradient estimator of the loss function with respect to $\vec{w_t}$, and $\vec{w_t}$ denotes the model's weights at iteration $t$.

**SGD Convergence Lemma.** The following lemma describes the convergence behavior for non-convex functions:

**Lemma A.1** (SGD Convergence for Non-Convex Functions). *Let $\ell(\vec{w})$ be a non-convex and $L$-smooth loss function, meaning that $\ell(\vec{w})$ has Lipschitz continuous gradients. Assume the learning rate $\eta_t = \eta$ is constant. Then, after $T$ iterations, the average squared gradient norm satisfies:*

$$\frac{1}{T}\sum_{t=0}^{T-1}\mathbb{E}\left[\|\nabla_{\vec{w}}\ell(\vec{w}_t)\|^2\right] \leq \frac{\ell(\vec{w}_0) - \ell(\vec{w}^*)}{T\eta} + \frac{L\eta\Gamma^2}{2} \tag{22}$$

Where $\ell(\vec{w}_0)$ is the initial loss, $\ell(\vec{w}^*)$ is the optimal (or minimum) loss, $L$ is the Lipschitz constant of the gradient (smoothness parameter), $\Gamma^2$ is the variance of the stochastic gradients, $T$ is the total number of iterations, and $\eta$ is the constant learning rate.

This lemma indicates that as $T$ increases, the expected norm of the gradient diminishes, meaning that the algorithm converges to a stationary point where the gradient is small. The convergence rate depends on the learning rate $\eta$ and the variance of the stochastic gradients.

# B CONVERGENCE IN INPUT GRADIENT NORM

**Theorem B.1** (Convergence in input gradient norm). *Let $k_g^t > 0$ be a constant relating the input and weight gradients, as established in Lemma D.1, and define $\kappa_t = (k_g^t)^2$ and $\Sigma_T = \sum_{t=0}^{T-1}(k_g^t)^2$. Then, under the SGD convergence assumptions, after $T$ iterations of SGD with a learning rate $\eta$, the following holds:*

$$\sum_{t=0}^{T-1}\|\nabla_X\ell(\vec{w}_t)\|_2^2 \leq \sum_{t=0}^{T-1}\frac{\kappa_t}{\eta}\mathbb{E}_t\left[\ell(\vec{w}_t) - \ell(\vec{w}_{t+1})\right] + \frac{\eta\mathcal{L}\Gamma^2\Sigma_T}{2} \tag{23}$$

**Sketch of Proof.** The proof of input gradient convergence assumes the loss function is $\mathcal{L}$-Lipschitz and the stochastic gradient estimates are unbiased. Thus, typical SGD analysis, e.g. Bubeck et al. (2015), can be used to upper bound the weight gradient's norm. Using a transformation lemma (Lemma D.1), the proof relates the weight gradient to the input gradient while incurring a scaling factor that depends on $k_g$. By telescoping the result over iterations, it follows that the cumulative input gradient is bounded by the cumulative loss decrease and a term depending on the learning rate and Lipschitz constant (see proof in Appendix E.1).

**Discussion.** This theorem establishes the convergence of the sample gradient norm when using stochastic gradient descent (SGD). It shows that the input gradient converges in a manner similar to the weight gradient, with an additional dependence on the scaling factor $\kappa_t$, which is determined by the weight norm. Moreover, the convergence is bounded by the suboptimality gap $\ell(\vec{w}_0) - \ell(\vec{w}_T)$, as well as by the Lipschitz continuity of the loss function and the learning rate $\eta$.

# C EXPERIMENTAL DETAILS

## C.1 SETUP

**Datasets.** We use CIFAR-10, CIFAR-100 (Krizhevsky et al., 2009) and ImageNet (Russakovsky et al., 2015) datasets. For experiments that use memorization scores, we use the pre-computed stability-based memorization scores from Feldman & Zhang (2020) which have been made publicly available by the authors.

**Architectures.** For all of experiments we train ResNet18 (He et al., 2016) models from scratch, expect for the cross architecture results in Table 4, where we train VGG16, MobileNetV2 and Inception (small inception as used by (Feldman & Zhang, 2020)). All the architectures used the same training recipe as described below.

**Training.** When training models on CIFAR-10 and CIFAR-100 the initial learning rate was set to 0.1 and trained for 200 epochs. The learning rate is decreased by 10 at epochs 120 and 180. When training on CIFAR-10 and CIFAR-100 datasets the batch size is set to 128. We use stochastic gradient descent for training with momentum set to 0.9 and weight decay set to 1e-4. For both

CIFAR-10 and CIFAR-100 datasets, we used the following sequence of data augmentations for training: resize ($32 \times 32$), random crop, and random horizontal flip, this is followed by normalization. For ImageNet we trained a ResNet18 for 200 epochs with the same setting except the resize random crop was set to $224 \times 224$.

**Testing.** During testing no augmentations were used, i.e. we used resize followed by normalization. To improve reproducibility, we have provided the code in the supplementary material.

## C.2 SIMILARITY WITH MEMORIZATION SCORES ACROSS ARCHITECTURES

**Experiment.** In this section, we present the results of measuring the cosine similarity between the proposed memorization proxies (CSL, CSG) and the memorization score from Feldman & Zhang (2020) across different architectures. Specifically, we tested VGG (Simonyan & Zisserman, 2014), MobileNetV2 (Sandler et al., 2018), and Inception (Szegedy et al., 2016).

**Results.** The results are shown in Table 4, which reports the cosine similarity between the CSL and CSG metrics and the memorization score for the three architectures on the CIFAR-100 dataset. These models were trained using the methodology described in Section C.1.

**Takeaways.** The results indicate that the top 5K (i.e., the similarity between the metrics and the top 5000 most memorized samples according to Feldman & Zhang (2020)) is highly consistent across architectures, and the overall match across the dataset is also quite high for CSL. However, two key observations are worth noting: (1) VGG16 shows a lower correlation on the 'All' category, and (2) CSG performs worse than CSL, similar to the findings for ResNet18 in the main paper.

| Architecture | Samples | CSL | CSG |
|---|---|---|---|
| VGG16 | Top 5K | 1.00 | 0.98 |
|  | All | 0.61 | 0.60 |
| MobileNetV2 | Top 5K | 0.95 | 0.94 |
|  | All | 0.73 | 0.65 |
| Inception | Top 5K | 0.97 | 0.95 |
|  | All | 0.70 | 0.64 |

Table 4: Cosine similarity between stability-based memorization score with CSL and CSG for different architectures on CIFAR-100 for Top 5K and all samples.

## C.3 COMPUTE COST ANALYSIS

In this section, we provide a detailed analysis of the computational cost of the proposed proxies in comparison to other techniques. We assume the cost of one forward pass through a neural network is $F$, and consequently, the cost of a backpropagation step is $2F$, making the total cost for one forward-backward pass $3F$. Using previously defined notation, let $m$ represent the dataset size and $T$ the total number of training epochs.

**Stability-Based Memorization.** Feldman & Zhang (2020) trained between 2,000 and 10,000 models to compute the stability-based memorization score. Thus, the total computational cost is $10,000 \cdot 3F \cdot T \cdot m$.

**Cumulative Sample Curvature.** Garg et al. (2024) trained a single model and proposed using sample curvature averaged over training to estimate memorization. Hutchinson's trace estimator was employed to calculate curvature, which requires 2 forward passes and 1 backward pass, repeated $n$ times over the entire dataset for each epoch. While their results show that $n$ ranges from 2 to 10, we use $n = 2$ to provide the computational advantage in their favor, even though $n = 10$ produces better results. Thus, the total cost consists of the training and curvature computation.

$$\text{Cost} = 3F \cdot T \cdot m + 4F \cdot T \cdot m \cdot n$$
$$= 3F \cdot T \cdot m + 4F \cdot T \cdot m \cdot 2$$
$$= 11F \cdot T \cdot m$$

If $n = 10$ is used for optimal results, as reported in Tables 2 and 3, the total computational cost becomes $43F \cdot T \cdot m$.

**CSL and CSG (Ours).** Both CSL and CSG can be obtained without additional computational cost during training. Therefore, the only cost is that of the training process, which is $3F \cdot T \cdot m$. The computational cost comparison is summarized in Table 5.

| Method | Absolute Cost | Relative Cost |
|---|---|---|
| Stability-Based (Feldman & Zhang, 2020) | $6000FTm - 30,000FTm$ | $2,000 \times - 30,000 \times$ |
| Cumulative Sample Curvature (Garg et al., 2024) | $11FTm - 43FTm$ | $3.6 \times - 14.33 \times$ |
| **CSL and CSG (Ours)** | $3FTm$ | $1 \times$ |

Table 5: Summary of the compute cost of the proposed metric compared to existing methods.

### C.4 Additional Details on Validating Theory Experiments

For the experiments described in Section 5.1, we provide additional details regarding the methodology. To generate the graphs in Figures 8, 9 and 10 we collected all relevant metrics for each sample in the dataset and grouped them into bins based on the x-axis metric in each figure. For instance, in Figure 8 samples were binned based on their learning time. Similarly, in Figure 9 , we binned samples based on their memorization scores as defined in Feldman & Zhang (2020) and and in Figure 10, the samples were again binned by learning time.

### C.5 Additional Details on Mislabelled Detection Experiments

In this section, we provide additional details regarding the setup for mislabel detection experiments. For all experiments, we trained the models using the training procedure outlined in Appendix C.1.

**Learning Time.** Learning time is defined as the first epoch at which the sample was correctly learned. For example, if a sample's correct predictions during training were $[0, 0, 0, 1, 1, 0, 1, 1, 0, 1, 1, 1]$, the learning time would be epoch 3.

**In Confidence.** In-confidence (Carlini et al., 2019a) is calculated as 1 - "the predicted probability" of the true class.

**Confident Learning.** For the implementation of confident learning (Northcutt et al., 2021), we utilized the cleanlab library, which is available at `https://github.com/cleanlab/cleanlab`. We applied 3-fold cross-validation to compute out-of-sample probability scores for the samples. These probability scores were then input into the cleanlab implementation to generate the results reported.

**SSFT.** Second Split Forgetting Time (SSFT) (Maini et al., 2022) is measured using two subsets, Set 1 and Set 2. A model is first trained on Set 1 and subsequently fine-tuned on Set 2, during which we measure how quickly a sample from Set 1 is misclassified or "forgotten". This process is repeated

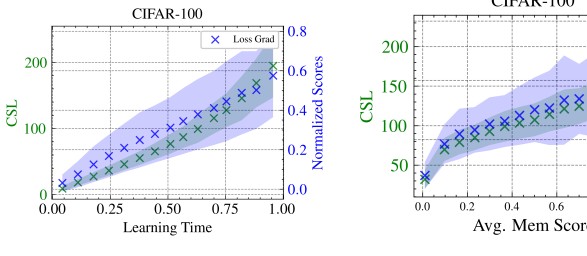

Figure 8: Learning time vs CSL and CSG on CIFAR-100 dataset.

Figure 9: Memorization score vs CSL and CSG on CIFAR-100 dataset.

Figure 10: Learning time vs memorization and input loss curvature on CIFAR-100.

Figure 11: Validating theoretical results of the proposed proxies on deep vison models trained on CIFAR-10 and CIFAR-100, with error bar.

for both subsets to cover the entire dataset. Specifically, after training on Set 1 and measuring the forgetting time for samples in Set 1 during fine-tuning on Set 2, the model is then trained on Set 2, and the forgetting time for Set 2 is measured during fine-tuning on Set 1.

**CSLT.** Cumulative Sample Learning Time (CSLT) can be considered as the cumulative sample loss, where the loss is a binary $0-1$ loss. For example, consider a sample's correct predictions during training over 12 epochs in this case. Let the correct predictions be $[0, 0, 0, 1, 1, 0, 1, 1, 0, 1, 1, 1]$. Then CSLT would be calculated as 'Total Epochs' - 'No. of ones' $= 12 - 7 = 5$, representing the total number of incorrect predictions (represented by 0s).

**Curvature.** To calculate the curvature of a sample, we used the technique described in Garg et al. (2024). The hyperparameters were set to $n = 10$ and $h = 0.001$, following the same configuration as outlined by Garg et al. (2024).

# D  RELATING INPUT AND WEIGHT GRADIENT, HESSIAN

## D.1  GRADIENT

We want to analyze the gradient of the loss function with respect to the input $X$, denoted as $\nabla_X \ell(X, Y)$, and relate it to the gradient with respect to the weights of the first layer, $\nabla_{W_1} \ell(X, Y)$, considering $X$ as a fat matrix (more features than samples a.k.a under determined).

**Lemma D.1** (Input gradient norm bound by weight gradient norm). *For any neural network with input $X$, the norm of the gradient of the loss function with respect to the input $X$ is bounded by the product of the norm of the gradient with respect to the weights of the first layer, the norm of the pseudo-inverse of $X$, and the norm of the weight matrix $W_1$. Formally,*

$$\|\nabla_X \ell(X, Y)\|_F \leq k_g \|\nabla_{W_1} \ell(X, Y)\|_F \tag{24}$$

*where $k_g = \dfrac{\|W_1^T\|_F \, \|(X^T)^+\|_F}{s_X}$.*

**Proof**:

$$\begin{aligned}
\nabla_X \ell(X, Y) &= \nabla_{\hat{Y}} \ell(X, Y) \, \nabla_H \hat{Y} \, \nabla_X H \\
&= W_1^T \, W_2^T \nabla_{\hat{Y}} \ell(X, Y) \odot \sigma'(H) \\
&= W_1^T \, W_2^T \nabla_{\hat{Y}} \ell(X, Y) \odot \sigma'(W_1 X) \tag{25}
\end{aligned}$$

Similarly for the gradient w.r.t to $W_1$ we have

$$\begin{aligned}
\nabla_{W_1} \ell(X, Y) &= \nabla_{\hat{Y}} \ell(X, Y) \, \nabla_H \hat{Y} \, \nabla_{W_1} H \\
&= W_2^T \nabla_{\hat{Y}} \ell(X, Y) \odot \sigma'(H) \, X^T \\
&= W_2^T \nabla_{\hat{Y}} \ell(X, Y) \odot \sigma'(W_1 X) \, X^T \tag{26}
\end{aligned}$$

From Equations 25 and 26 we have:

$$W_1^T \nabla_{W_1} \ell(X, Y) = \nabla_X \ell(X, Y) X^T \tag{27}$$

$$W_1^T \nabla_{W_1} \ell(X, Y)(X^T)^+ = \nabla_X \ell(X, Y) X^T (X^T)^+$$

$$\|W_1^T \nabla_{W_1} \ell(X, Y)(X^T)^+\|_F = \|\nabla_X \ell(X, Y) X^T (X^T)^+\|_F$$

Let $s_X$ denote the smallest singular value of $P = X^T (X^T)^+$

$$\|W_1^T\|_F \|\nabla_{W_1} \ell(X, Y)\|_F \|(X^T)^+\|_F \geq \|\nabla_X \ell(X, Y) X^T (X^T)^+\|_F$$

$$\|W_1^T\|_F \|\nabla_{W_1} \ell(X, Y)\|_F \|(X^T)^+\|_F \geq \|\nabla_X \ell(X, Y) X^T (X^T)^+\|_F \geq s_X \|\nabla_X \ell(X, Y)\|_F$$

Thus we have

$$\|\nabla_X \ell(X, Y)\|_F \leq \frac{\|W_1^T\|_F \, \|(X^T)^+\|_F}{s_X} \|\nabla_{W_1} \ell(X, Y)\|_F$$

$$\|\nabla_X \ell(X, Y)\|_F \leq k_g \|\nabla_{W_1} \ell(X, Y)\|_F \quad \blacksquare$$

## D.2 Hessian

**Lemma D.2** (Input Hessian norm bound by weight Hessian norm). *If the assumption of $\rho$-Lipschitz of the loss function holds, then for any neural network with input $X$, the norm of the Hessian of the loss function with respect to the input $X$ is bounded by the norm of the Hessian with respect to the weights of the first layer, the norm of the pseudo-inverse of $X$, and the norms of the weight matrices. Formally,*

$$\|\nabla^2_{W_1}\ell(X,Y)\|_F \leq k_h \|\nabla^2_X \ell(X,Y)\|_F \tag{28}$$

*where $k_h = \dfrac{1}{s_{W_1} s_{W_1^+}} \|(W_1^T)^+\|_F^2 \, \|X^T\|_F^2$*

**Proof**

$$\nabla_X \ell(X,Y) = W_1^T \, W_2^T \nabla_{\hat{Y}} \ell(X,Y) \odot \sigma'(W_1 X)$$

We rewrite this using Einstein notation since the Hessian that we will be dealing with are 4D tensor.

$$(\nabla_X \ell(X,Y))_{ij} = (W_1^T)_{ik}(W_2^T)_{kl}(\nabla_{\hat{Y}}\ell(X,Y))_{lj}\sigma'(W_1 X)_{kj}$$

Now we consider $\nabla^2_X$

$$
\begin{aligned}
\nabla^2_X \ell(X,Y) &= W_1^T \left( W_2^T \nabla_X \nabla_{\hat{Y}} \ell(X,Y) \odot \sigma'(W_1 X) \right) \\
&\quad + W_1^T \left( W_1^T \, W_2^T \nabla_{\hat{Y}} \ell(X,Y) \odot \sigma''(W_1 X) \right) \\
&= W_1^T \left( W_2^T \nabla_H \nabla_{\hat{Y}} \ell(X,Y) \nabla_X H \odot \sigma'(W_1 X) \right) \\
&\quad + W_1^T \left( W_1^T \, W_2^T \nabla_{\hat{Y}} \ell(X,Y) \odot \sigma''(W_1 X) \right) \\
&= T_1 + T_2
\end{aligned}
\tag{29}
$$

Rewriting in Einstein Notation we have

$$(T_1)_{ijkl} = (W_1^T)_{im}(W_1^T)_{jn}(W_2^T)_{ko}(\nabla_H \nabla_{\hat{Y}}\ell(X,Y))_{mnop}\sigma'(W_1 X)_{pl}$$
$$(T_2)_{ijkl} = (W_1^T)_{im}(W_1^T)_{jn}(W_1^T)_{ql}(W_2^T)_{ko}(\nabla_{\hat{Y}}\ell(X,Y))_{mnop}\sigma''(W_1 X)_{pq}$$

Now we consider $\nabla^2_{W_1}$

$$
\begin{aligned}
\nabla^2_{W_1} \ell(X,Y) &= \left( W_2^T \nabla_{W_1} \nabla_{\hat{Y}} \ell(X,Y) \odot \sigma'(W_1 X) \right) X^T \\
&\quad + \left( W_2^T \nabla_{\hat{Y}} \ell(X,Y) \odot \left( \sigma''(W_1 X) X^T \right) \right) X^T \\
&= \left( W_2^T \nabla_H \nabla_{\hat{Y}} \ell(X,Y) \nabla_{W_1} H \odot \sigma'(W_1 X) \right) X^T \\
&\quad + \left( W_2^T \nabla_{\hat{Y}} \ell(X,Y) \odot \left( \sigma''(W_1 X) X^T \right) \right) X^T \\
&= \left( W_2^T \nabla_H \nabla_{\hat{Y}} \ell(X,Y) X^T \odot \sigma'(W_1 X) \right) X^T \\
&\quad + \left( W_2^T \nabla_{\hat{Y}} \ell(X,Y) \odot \left( \sigma''(W_1 X) X^T \right) \right) X^T \\
&= \left( W_2^T \nabla_H \nabla_{\hat{Y}} \ell(X,Y) \odot \sigma'(W_1 X) \right) X^T X^T \\
&\quad + \left( W_2^T \nabla_{\hat{Y}} \ell(X,Y) \odot \left( \sigma''(W_1 X) X^T \right) \right) X^T \\
&= T_3 + T_4
\end{aligned}
\tag{30}
$$

Rewriting in Einstein Notation we have

$$(T_3)_{ijkl} = (W_2^T)_{ko}(\nabla_H \nabla_{\hat{Y}}\ell(X,Y))_{mnop}\sigma'(W_1 X)_{pq}(X^T)_{ql}(X^T)_{ij}$$
$$(T_4)_{ijkl} = (W_2^T)_{ko}(\nabla_{\hat{Y}}\ell(X,Y))_{mnop}\sigma''(W_1 X)_{pq}(X^T)_{ql}(X^T)_{ij}$$

Using Equations 29 and 30 we have

$$(W_1^T)_{ia}(W_1^T)_{jb}(\nabla^2_{W_1}\ell(X,Y))_{abkl} = (\nabla^2_X \ell(X,Y))_{ijkl}(X^T)_{km}(X^T)_{ln}$$
$$W_1^T W_1^T \nabla^2_{W_1}\ell(X,Y) = \nabla^2_X \ell(X,Y) X^T X^T$$
$$(W_1^T)^+(W_1^T)^+ W_1^T W_1^T \nabla^2_{W_1}\ell(X,Y) = (W_1^T)^+(W_1^T)^+ \nabla^2_X \ell(X,Y) X^T X^T$$

Note the two $(W_1^T)^+$ are across different axes

$$\|(W_1^T)^+(W_1^T)^+W_1^TW_1^T\nabla^2_{W_1}\ell(X,Y)\|_F = \|(W_1^T)^+(W_1^T)^+\nabla^2_X\ell(X,Y)X^TX^T\|_F$$

$$\|(W_1^T)^+(W_1^T)^+W_1^TW_1^T\nabla^2_{W_1}\ell(X,Y)\|_F \leq \|(W_1^T)^+(W_1^T)^+\|_F\|\nabla^2_X\ell(X,Y)\|_F\|X^TX^T\|_F$$

$$s_{W_1}s_{W_1^+}\|\nabla^2_{W_1}\ell(X,Y)\|_F \leq k'_h\|\nabla^2_X\ell(X,Y)\|_F$$

$$\|\nabla^2_{W_1}\ell(X,Y)\|_F \leq \frac{k'_h}{s_{W_1}s_{W_1^+}}\|\nabla^2_X\ell(X,Y)\|_F$$

$$\|\nabla^2_{W_1}\ell(X,Y)\|_F \leq k_h\|\nabla^2_X\ell(X,Y)\|_F \quad \blacksquare$$

# E  LEARNING DYNAMICS

## E.1  INPUT GRADIENT CONVERGENCE

**Proof of Theorem B.1**

**Assumptions**:

- Assume that the loss function $\ell$ is $\mathcal{L}$-Lipschitz.
- Assume that the gradient estimator $\widetilde{\nabla}_w \ell$ is unbiased i.e., $\mathbb{E}_t[\widetilde{\nabla}_w \ell(\vec{w}_t)] = \nabla_w \ell(\vec{w}_t)$
- Assume bounded gradient as stated in Assumption 18.

**Proof**

Let $\vec{w}_t^{\,1}$ denote the vector of the first-layer weight parameters at the $t^{th}$ iteration. For simplicity, we slightly abuse the notation by using $\vec{w}_t = \vec{w}_t^{\,1}$ and $\nabla_w = \nabla_{\vec{w}_t}$ throughout the paper for ease of reference. Using the $\mathcal{L}$-Lipschitz assumption on the loss, for any vectors $\vec{w}_t$ and $\vec{w}_{t+1}$, the function $\ell$ satisfies the quadratic upper bound:

$$\ell(\vec{w}_{t+1}) \leq \ell(\vec{w}_t) + \langle \nabla_w \ell(\vec{w}_t), \vec{w}_{t+1} - \vec{w}_t \rangle + \frac{\mathcal{L}}{2} \|\vec{w}_t - \vec{w}_{t+1}\|_2^2$$

Based on the assumptions, the stochastic gradient descent (SGD) lemma provides the following result for two consecutive iterates $\vec{w}_t$ and $\vec{w}_{t+1}$ produced by the SGD algorithm:

$$\mathbb{E}_t[\ell(\vec{w}_{t+1})] \leq \ell(\vec{w}_t) - \eta \|\nabla_{\vec{w}} \ell(\vec{w}_t)\|_2^2 + \frac{\mathcal{L}}{2} \eta^2 \mathbb{E}_t[\|\widetilde{\nabla}_{\vec{w}} \ell(\vec{w}_t)\|_2^2]$$

where $\eta > 0$ is the step size of the algorithm. For stochastic gradient descent with learning rate $\eta$ and bounded gradient as defined in Assumption 18 we a have

$$\|\nabla_w \ell(\vec{w}_t)\|_2^2 \leq \frac{1}{\eta}(\mathbb{E}_t[\ell(\vec{w}_t) - \ell(\vec{w}_{t+1})]) + \frac{\eta \mathcal{L}\Gamma^2}{2}$$

$$(k_g^t)^2 \|\nabla_w \ell(\vec{w}_t)\|_2^2 \leq (k_g^t)^2 \frac{1}{\eta}(\mathbb{E}_t[\ell(\vec{w}_t) - \ell(\vec{w}_{t+1})]) + \frac{(k_g^t)^2 \eta \mathcal{L}\Gamma^2}{2} \qquad \text{Multiply by } (k_g^t)^2$$

$$\|\nabla_X \ell(\vec{w}_t)\|_2^2 \leq (k_g^t)^2 \frac{1}{\eta}(\mathbb{E}_t[\ell(\vec{w}_t) - \ell(\vec{w}_{t+1})]) + \frac{(k_g^t)^2 \eta \mathcal{L}\Gamma^2}{2} \qquad \text{Using Lemma D.1}$$

Telescoping the result for $T$ iterations

$$\sum_{t=0}^{T-1} \|\nabla_X \ell(\vec{w}_t)\|_2^2 \leq \frac{1}{\eta}\left(\sum_{t=0}^{T-1} (k_g^t)^2 \mathbb{E}_t[\ell(\vec{w}_t) - \ell(\vec{w}_{t+1})]\right) + \frac{\eta \mathcal{L}\Gamma^2 \sum_{t=0}^{T-1} (k_g^t)^2}{2}$$

Let $(k_g^t)^2 = \kappa_t$ and $\Sigma_T = \sum_{t=0}^{T-1} (k_g^t)^2$, then we have the result

$$\sum_{t=0}^{T-1} \|\nabla_X \ell(\vec{w}_t)\|_2^2 \leq \sum_{t=0}^{T-1} \frac{\kappa_t}{\eta} \mathbb{E}_t[\ell(\vec{w}_t) - \ell(\vec{w}_{t+1})] + \frac{\eta \mathcal{L}\Gamma^2 \Sigma_T}{2} \qquad \blacksquare$$

## E.2  LEARNING TIME AND MEMORIZATION

**Proof of Theorem 4.1**

**Assumptions**:

- Assume that the loss function $\ell$ is $\mathcal{L}$-Lipschitz.
- Assume that the gradient estimator $\widetilde{\nabla}_w \ell$ is unbiased i.e., $\mathbb{E}_t[\widetilde{\nabla}_w \ell(\vec{w}_t)] = \nabla_w \ell(\vec{w}_t)$
- Assume bounded gradient as stated in Assumption 18.
- Assume loss is bounded $0 \leq \ell \leq L$
- Assume loss variance is $\sigma_l$

**Proof**

Using the sample learning condition from Equation 4 we have:

$$\frac{1}{T} \sum_{t=0}^{T-1} \|\nabla_X \ell(\vec{w}_t)\|_2^2 \leq \tau$$

For our analysis it is easier to analyze the results with a minor modification to Theorem B.1 by setting $\kappa_m = \max_{t \in \{1, \cdots, T\}} \kappa_t$. Then we have

$$\sum_{t=0}^{T-1} \|\nabla_X \ell(\vec{w}_t)\|_2^2 \leq \frac{\kappa_m}{\eta} \sum_{t=0}^{T-1} \mathbb{E}_t \left[ \ell(\vec{w}_t) - \ell(\vec{w}_{t+1}) \right] + \frac{\kappa_m \eta L \Gamma^2 T}{2}$$

$$\frac{1}{T} \sum_{t=0}^{T-1} \|\nabla_X \ell(\vec{w}_t)\|_2^2 \leq \frac{\kappa_m}{T\eta} \sum_{t=0}^{T-1} \mathbb{E}_t \left[ \ell(\vec{w}_t) - \ell(\vec{w}_{t+1}) \right] + \frac{\kappa_m \eta L \Gamma^2}{2} \tag{31}$$

The threshold for learning a sample from Equation 31 is given by

$$\tau = \frac{\kappa_m}{T\eta} \sum_{t=0}^{T-1} \mathbb{E}_t \left[ \ell(\vec{w}_t) - \ell(\vec{w}_{t+1}) \right] + \frac{\kappa_m \eta L \Gamma^2}{2}$$

Consider two samples $a, b$ then for both $a$ and $b$ to be learnt, let $T_a$ and $T_b$ represent the learning time for $a$ and $b$ respectively. Then if both samples are learnt $\tau_a = \tau_b \implies$

$$\tau_a = \tau_b \tag{32}$$

$$\frac{\kappa_m}{T_a \eta} \sum_{t=0}^{T_a-1} \mathbb{E}_t \left[ \ell(\vec{w}_t) - \ell(\vec{w}_{t+1}) \right] + \frac{\kappa_m \eta L \Gamma^2}{2} = \frac{\kappa_m}{T_b \eta} \sum_{t=0}^{T_b-1} \mathbb{E}_t \left[ \ell(\vec{w}_t) - \ell(\vec{w}_{t+1}) \right] + \frac{\kappa_m \eta L \Gamma^2}{2}$$

$$\frac{\kappa_m \sum_{t=0}^{T_a-1} \mathbb{E}_t \left[ \ell(\vec{w}_t) - \ell(\vec{w}_{t+1}) \right]}{T_a \eta} = \frac{\kappa_m \sum_{t=0}^{T_b-1} \mathbb{E}_t \left[ \ell(\vec{w}_t) - \ell(\vec{w}_{t+1}) \right]}{T_b \eta}$$

$$T_b \sum_{t=0}^{T_a} \mathbb{E}_t \left[ \ell(\vec{w}_t) - \ell(\vec{w}_{t+1}) \right] = T_a \sum_{t=0}^{T_b} \mathbb{E}_t \left[ \ell(\vec{w}_t) - \ell(\vec{w}_{t+1}) \right]$$

$$T_b \mathbb{E}_t \left[ \ell(\vec{w}_{T_a}) - \ell(\vec{w}_0) \right] = T_a \mathbb{E}_t \left[ \ell(\vec{w}_{T_b}) - \ell(\vec{w}_0) \right] \tag{33}$$

$$\mathbb{E}_t \left[ \ell(\vec{w}_0) \right] (T_a - T_b) = T_a \mathbb{E}_t \left[ \ell(\vec{w}_{T_b}) \right] - T_b \mathbb{E}_t \left[ \ell(\vec{w}_{T_a}) \right] \tag{34}$$

Let $\ell^{\backslash a}$ denote the loss of the model when $a$ was removed from the training set i.e. the training set is $S^{\backslash a}$. Let $\vec{w}^*$ denote the optimal for $S^{\backslash a}$. If we add and subtract this from Equation 34 we have

$$\mathbb{E}_t \left[ \ell(\vec{w}_0) \right] (T_a - T_b) = T_a \mathbb{E}_t \left[ \ell(\vec{w}_{T_b}) - \ell^{\backslash b}(\vec{w}^*) \right] - T_b \mathbb{E}_t \left[ \ell(\vec{w}_{T_a}) - \ell^{\backslash a}(\vec{w}^*) \right]$$
$$- T_a \ell^{\backslash b}(\vec{w}^*) + T_b \ell^{\backslash a}(\vec{w}^*)$$

$$\mathbb{E}_t \left[ \ell(\vec{w}_0) \right] (T_a - T_b) = T_a \mathbb{E}_t \left[ \ell(\vec{w}_{T_b}) - \ell^{\backslash b}(\vec{w}^*) \right] - T_b \mathbb{E}_t \left[ \ell(\vec{w}_{T_a}) - \ell^{\backslash a}(\vec{w}^*) \right]$$
$$- T_a \ell^{\backslash b}(\vec{w}^*) + T_b \ell^{\backslash a}(\vec{w}^*)$$

Now we take the expectation over the randomness of the algorithm $\phi \sim \mathbf{\Phi}$.

$$\mathbb{E}_\phi \left[ \ell(\vec{w}_0)(T_a - T_b) \right] = \mathbb{E}_\phi \left[ T_a \right] \mathbb{E}_\phi \left[ \ell(\vec{w}_{T_b}) - \ell^{\backslash b}(\vec{w}^*) \right] - \mathbb{E}_\phi \left[ T_b \right] \mathbb{E}_{t,\phi} \left[ \ell(\vec{w}_{T_a}) - \ell^{\backslash a}(\vec{w}^*) \right]$$
$$- \mathbb{E}_\phi \left[ T_a \ell^{\backslash b}(\vec{w}^*) + T_b \ell^{\backslash a}(\vec{w}^*) \right]$$

Observe that

$$\mathbb{E}_{t,\phi} \left[ \ell(\vec{w}_{T_b}) - \ell^{\backslash b}(\vec{w}^*) \right] = \mathbb{E}_\phi \left[ \ell(\vec{w}_{T_b}) - \ell^{\backslash b}(\vec{w}^*) \right]$$

since the term does not depend on $t$. Furthermore, note that $\mathbb{E}_\phi \left[ \ell(\vec{w}_{T_b}) - \ell^{\backslash b}(\vec{w}^*) \right]$ corresponds to the memorization metric as defined by Feldman (2020), expressed in terms of loss, see proof in

A.3 from Ravikumar et al. (2024a). We denote $\mathbb{E}_\phi[T_a] = \hat{T}_a$. To ensure the correct sign of the memorization score, we assume $\ell^{\backslash b}(\vec{w}^*) \geq \ell(\vec{w}_{T_b})$, as optimizing the loss for a data point included in the training set should result in a loss that is lower than or equal to the loss when the data point is excluded. Thus $\ell(\vec{w}_{T_b}) - \ell^{\backslash b}(\vec{w}^*) \leq 0$, since $\mathrm{mem}(b) \geq 0$, we must adjust the sign accordingly. Additionally the losses are for all the samples. To convert them to memorization scores we assume that the loss can be estimated using the sample of interest with a variance $\sigma_l$. Thus using Chebyshev's inequality we have can state with a confidence $1 - \delta$:

$$\mathbb{E}_\phi\left[\ell(\vec{w}_0)(T_a - T_b)\right] \leq -\hat{T}_a L \,\mathrm{mem}(b) - \left(-\hat{T}_b L \,\mathrm{mem}(a)\right) + \mathbb{E}_\phi\left[-T_a \ell^{\backslash b}(w^*) + T_b \ell^{\backslash a}(w^*)\right]$$
$$+ \frac{2T_{max}\sigma_l}{\sqrt{\delta}}$$
$$\mathbb{E}_\phi\left[\ell(\vec{w}_0)(T_a - T_b)\right] \leq -\hat{T}_a L \,\mathrm{mem}(b) + \hat{T}_b L \,\mathrm{mem}(a) + \mathbb{E}_\phi\left[T_b \ell^{\backslash a}(\vec{w}^*) - T_a \ell^{\backslash b}(\vec{w}^*)\right]$$
$$+ \frac{2T_{max}\sigma_l}{\sqrt{\delta}} \mathbb{E}_\phi\left[\ell(\vec{w}_0)(T_a - T_b)\right]$$
$$\leq \hat{T}_b L \,\mathrm{mem}(a) - \hat{T}_a L \,\mathrm{mem}(b) + \mathbb{E}_\phi\left[T_a \ell^{\backslash a}(\vec{w}^*) - T_a \ell^{\backslash b}(\vec{w}^*)\right] + \frac{2T_{max}\sigma_l}{\sqrt{\delta}}$$
$$\text{Assume } T_a > T_b$$
$$\mathbb{E}_\phi\left[\ell(\vec{w}_0)(T_a - T_b)\right] \leq \hat{T}_b L \,\mathrm{mem}(a) - \hat{T}_a L \,\mathrm{mem}(b) + \mathbb{E}_\phi\left[T_a \ell(\vec{w}^*) - T_a \ell(\vec{w}^*) + 2T_a\beta\right] + \frac{2T_{max}\sigma_l}{\sqrt{\delta}}$$
$$\mathbb{E}_\phi\left[\ell(\vec{w}_0)(T_a - T_b)\right] \leq 2T_{max} L \,\mathrm{mem}(a) + 2T_{max}\beta + \frac{2\sigma_l}{\sqrt{\delta}}$$
$$\mathbb{E}_\phi\left[\ell(\vec{w}_0)(T_a - T_{ref})\right] \leq T_{max} L \,\mathrm{mem}(a) + T_{max}\left(2\beta + \frac{2\sigma_l}{\sqrt{\delta}}\right)$$
$$\mathbb{E}_\phi\left[T_a - T_{ref}\right] \leq \frac{T_{max}L}{\mathbb{E}_\phi\left[\ell(\vec{w}_0)\right]}\left(\mathrm{mem}(a) + \frac{2\beta}{L} + \frac{2\sigma_l}{L\sqrt{\delta}}\right)$$
$$\mathbb{E}_\phi\left[T_a\right] - \hat{T}_{ref} \leq \frac{T_{max}L}{\mathbb{E}_\phi\left[\ell(\vec{w}_0)\right]}\left(\mathrm{mem}(a) + \frac{2\beta}{L} + \frac{2\sigma_l}{L\sqrt{\delta}}\right) \quad \blacksquare$$

### E.3 Learning Time and Mean Sample Loss

**Proof of Theorem 4.2**

**Assumptions**:

- Assume that the loss function $\ell$ is $\mathcal{L}$-Lipschitz.
- Assume that the gradient estimator $\widetilde{\nabla}_w \ell$ is unbiased i.e., $\mathbb{E}_t[\widetilde{\nabla}_w \ell(\vec{w}_t)] = \nabla_w \ell(\vec{w}_t)$
- Assume bounded gradient as stated in Assumption 18.
- Assume for some $\lambda$, $\ell(\vec{w}_{T_{ref}}) = (1 - \lambda)\ell(\vec{w}_0)$.
- Assume loss variance is $\sigma_l$

**Proof**

We assume a trivial lower bound on $\mathbb{E}[\ell(\vec{w}_t)] \geq \ell(\vec{w}^*)$

Consider Equation 33

$$T_{z_a} \mathbb{E}_t\left[\ell(\vec{w}_0) - \ell(\vec{w}_{T_{z_b}})\right] = T_{z_b} \mathbb{E}_t\left[\ell(\vec{w}_0) - \ell(\vec{w}_{T_{z_a}})\right]$$

Let $T_b = T_{ref}$

$$T_{z_a} \lambda \mathbb{E}_t\left[\ell(\vec{w}_0)\right] = T_{ref} \mathbb{E}_t\left[\ell(\vec{w}_0) - \ell(\vec{w}_{T_{z_a}})\right]$$

Telescope the term on the right

$$T_{ref}\, \mathbb{E}\left[\ell(\vec{w}_0) - \ell(\vec{w}_{T_{z_a}})\right] = T_{ref} \sum_{t=0}^{T_{z_a}-1} \mathbb{E}\left[\ell(\vec{w}_t) - \ell(\vec{w}_{t+1})\right]$$

We split the sum into two parts:

$$T_{ref} \sum_{t=0}^{T_{z_a}-1} \mathbb{E}_t\left[\ell(\vec{w}_t) - \ell(\vec{w}_{t+1})\right] = T_{ref}\left(\sum_{t=0}^{T_{z_a}-1} \mathbb{E}_t\left[\ell(\vec{w}_t)\right] - \sum_{t=0}^{T_{z_a}-1} \mathbb{E}\left[\ell(\vec{w}_{t+1})\right]\right)$$

Using the lower bound on $\mathbb{E}_t[\ell(\vec{w}_{t+1})]$ we have

$$T_{ref} \sum_{t=0}^{T_{z_a}-1} \mathbb{E}_t\left[\ell(\vec{w}_t) - \ell(\vec{w}_{t+1})\right] \leq T_{ref}\left(\sum_{t=0}^{T_{z_a}-1} \mathbb{E}_t\left[\ell(\vec{w}_t)\right] - T_{z_a}\ell(\vec{w}^*)\right)$$

Using this in Equation 33 we get

$$T_{z_a}\lambda\, \mathbb{E}_t\left[\ell(\vec{w}_0)\right] \leq T_{ref}\left(\sum_{t=0}^{T_{z_a}-1} \mathbb{E}\left[\ell(\vec{w}_t)\right] - T_{z_a}\ell(\vec{w}^*)\right)$$

Grouping $T_{z_a}$ terms we have

$$T_{z_a}\left(\lambda\, \mathbb{E}_t\left[\ell(\vec{w}_0)\right] + T_{ref}\ell(\vec{w}^*)\right) \leq T_{ref} \sum_{t=0}^{T_{z_a}-1} \mathbb{E}_t\left[\ell(\vec{w}_t)\right]$$

We can estimate $\ell(\vec{w}_t)$ using $\mathbb{E}_{z_i}[\ell(\vec{w}_t, z_i)]$. Assume the variance of the loss estimate is $\sigma_l$. By Chebyshev's inequality, for any $\delta > 0$, with probability at least $1 - \delta$:

$$\ell(\vec{w}_t) \leq \ell(\vec{w}_t, \vec{z}_i) + \frac{\sigma_l}{\sqrt{\delta}}$$

Using this we have

$$T_{z_a}\left(\lambda\, \mathbb{E}_t\left[\ell(\vec{w}_0)\right] + T_{ref}\ell(\vec{w}^*)\right) \leq T_{ref} \sum_{t=0}^{T_{z_a}-1}\left[\ell(\vec{w}_t, \vec{z}_a) + \frac{\sigma_l}{\sqrt{\delta}}\right]$$

$$T_{z_a}\left(\lambda\, \mathbb{E}_t\left[\ell(\vec{w}_0)\right] + T_{ref}\ell(\vec{w}^*)\right) \leq T_{ref} \sum_{t=0}^{T_{z_a}-1}\left[\ell(\vec{w}_t, \vec{z}_a)\right] + T_{ref}T_{z_a}\frac{\sigma_l}{\sqrt{\delta}}$$

$$T_{z_a}\left(\lambda\, \mathbb{E}_t\left[\ell(\vec{w}_0)\right] + T_{ref}\ell(\vec{w}^*)\right) \leq T_{ref}\, \mathrm{CSL}(\vec{z}_a) + T_{ref}T_{z_a}\frac{\sigma_l}{\sqrt{\delta}}$$

$$T_{z_a}\left(\lambda\, \mathbb{E}_t\left[\ell(\vec{w}_0)\right] + T_{ref}\ell(\vec{w}^*) - T_{ref}\frac{\sigma_l}{\sqrt{\delta}}\right) \leq T_{ref}\, \mathrm{CSL}(\vec{z}_a)$$

Thus we have

$$T_{z_a} \leq \frac{T_{ref}\, \mathrm{CSL}(\vec{z}_a)}{\left(\lambda\, \mathbb{E}_t\left[\ell(\vec{w}_0)\right] + T_{ref}\ell(\vec{w}^*) - T_{ref}\dfrac{\sigma_l}{\sqrt{\delta}}\right)}$$

$$T_{z_a} \leq \frac{\mathrm{CSL}(\vec{z}_a)}{\dfrac{\lambda\ell(\vec{w}_0)}{T_{ref}} + \ell(\vec{w}^*) - \dfrac{\sigma_l}{\sqrt{\delta}}} \quad \blacksquare$$

### E.4 GRADIENT AND LEARNING TIME

**Proof of Theorem 4.3**

**Assumptions**:

- Assume that the loss function $\ell$ is $\mathcal{L}$-Lipschitz.
- Assume that the gradient estimator $\widetilde{\nabla}_w \ell$ is unbiased i.e., $\mathbb{E}_t[\widetilde{\nabla}_w \ell(\vec{w}_t)] = \nabla_w \ell(\vec{w}_t)$.
- Assume bounded gradient as stated in Assumption 18.
- Assume for some $\lambda$, $\ell(\vec{w}_{T_{ref}}) = (1 - \lambda)\ell(\vec{w}_0)$.
- Assume the loss $\ell$ satisfies $\mu$-PL condition.

**Proof**

Using Equation 27 we have:

$$W_1^T \nabla_{W_1} \ell(X, Y) = \nabla_X \ell(X, Y) X^T$$
$$(W_1^T)^+ W_1^T \nabla_{W_1} \ell(X, Y) = (W_1^T)^+ \nabla_X \ell(X, Y) X^T$$
$$\|(W_1^T)^+ W_1^T \nabla_{W_1} \ell(X, Y)\|_F = \|(W_1^T)^+ \nabla_X \ell(X, Y) X^T\|_F$$
$$\|(W_1^T)^+ W_1^T \nabla_{W_1} \ell(X, Y)\|_F \leq \|(W_1^T)^+\|_F \|\nabla_X \ell(X, Y)\|_F \|X^T\|_F$$
$$s_W \|\nabla_{W_1} \ell(X, Y)\|_F \leq \|(W_1^T)^+\|_F \|\nabla_X \ell(X, Y)\|_F \|X^T\|_F$$

Let $s_W$ be the smallest singular value of $(W_1^T)^+ W_1^T$

$$s_W \|\nabla_{W_1} \ell(X, Y)\|_F \leq k'_{gw} \|\nabla_X \ell(X, Y)\|_F \tag{35}$$
$$\|\nabla_{W_1} \ell(X, Y)\|_F \leq k_{gw} \|\nabla_X \ell(X, Y)\|_F \tag{36}$$

Let $\ell$ satisfy $\mu$-PL condition then we have:

$$\frac{1}{2}\|\nabla_{W_1} \ell(\vec{w})\|_2^2 \geq \mu \left[\ell(\vec{w}) - \ell(\vec{w}_*)\right]$$

$$k_{gw}^2 \frac{1}{2}\|\nabla_{W_1} \ell(\vec{w})\|_2^2 \geq k_{gw}^2 \mu \left[\ell(\vec{w}) - \ell(\vec{w}_*)\right]$$

$$\frac{1}{2}\|\nabla_X \ell(\vec{w})\|_2^2 \geq \mathcal{M} \left[\ell(\vec{w}) - \ell(\vec{w}_*)\right] \tag{37}$$

Now summing this over time we have

$$\frac{1}{2}\sum_{t=0}^{T-1} \|\nabla_X \ell(w_t)\|_2^2 \geq \mathcal{M} \sum_{t=0}^{T-1} \left[\ell(w_t) - \ell(w_*)\right] \tag{38}$$

Now consider Equation 33

$$T_a \, \mathbb{E}_t \left[\ell(\vec{w}_0) - \ell(\vec{w}_{T_b})\right] = T_b \, \mathbb{E}_t \left[\ell(\vec{w}_0) - \ell(\vec{w}_{T_a})\right]$$

Telescope the term on the right

$$T_a \, \mathbb{E}_t \left[\ell(\vec{w}_0) - \ell(\vec{w}_{T_b})\right] = T_b \sum_{t=0}^{T_a - 1} \mathbb{E}_t \left[\ell(\vec{w}_t) - \ell(\vec{w}_{t+1})\right]$$

$$T_a \, \mathbb{E}_t \left[\ell(\vec{w}_0) - \ell(\vec{w}_{T_b})\right] \leq T_b \sum_{t=0}^{T_a - 1} \mathbb{E}_t \left[\ell(\vec{w}_t) - \ell(\vec{w}_*)\right] \tag{39}$$

$$\leq \frac{\mathcal{M} T_b}{2} \sum_{t=0}^{T_a - 1} \|\nabla_X \ell(\vec{w}_t)\|_2^2$$

$$\leq \frac{\mathcal{M} T_{max}}{2} \sum_{t=0}^{T_a - 1} \|\nabla_X \ell(\vec{w}_t)\|_2^2$$

Consider a reference sample $T_{ref} = T_b$, then

$$T_a \, \mathbb{E}_t \left[\ell(\vec{w}_0) - \ell(\vec{w}_{T_b})\right] \leq \frac{\mathcal{M} T_{max}}{2} \sum_{t=0}^{T_a - 1} \|\nabla_X \ell(\vec{w}_t)\|_2^2$$

$$T_a \leq \frac{\mathcal{M} T_{max}}{2 \, \mathbb{E}_t \left[\ell(\vec{w}_0) - \ell(\vec{w}_{T_{ref}})\right]} \sum_{t=0}^{T_a - 1} \|\nabla_X \ell(\vec{w}_t)\|_2^2$$

Assume for some $\lambda$, $\ell(\vec{w}_{T_{ref}}) = (1-\lambda)\ell(\vec{w}_0)$ is true, then we have

$$T_a \leq \frac{\mathcal{M}T_{max}}{2\lambda\,\mathbb{E}_t\left[\ell(\vec{w}_0)\right]} \sum_{t=0}^{T_a-1} \|\nabla_X \ell(\vec{w}_t)\|_2^2$$

$$T_a \leq \frac{\mathcal{M}T_{max}}{2\lambda\ell(\vec{w}_0)} \sum_{t=0}^{T_{max}-1} \|\nabla_X \ell(\vec{w}_t)\|_2^2 \quad \blacksquare$$

### E.5 TRAINING DYNAMICS OF CURVATURE

**Proof of Theorem 4.6**

**Assumptions:**

- Assume that the loss function $\ell$ is $\mathcal{L}$-Lipschitz.
- Assume that the Hessian is $\rho$-Lipschitz.
- Assume that the gradient estimator $\widetilde{\nabla}_w \ell$ is unbiased i.e., $\mathbb{E}_t[\widetilde{\nabla}_w \ell(\vec{w}_t)] = \nabla_w \ell(\vec{w}_t)$.
- Assume bounded gradient as stated in Assumption 18.
- Assume for some $\lambda$, $\ell(\vec{w}_{T_{ref}}) = (1-\lambda)\ell(\vec{w}_0)$.
- Assume the loss $\ell$ satisfies $\mu$-PL condition.
- Assume bounded gradient variance $\mathbb{E}_t[\|\tilde{\nabla}_{w_1}\ell(\vec{w_1}) - \nabla_{w_1}\ell(\vec{w_1})\|^2] \leq \sigma^2$

**Proof**

If Lipschitz assumption 17 on the Hessian of $\ell$ holds from Nesterov & Polyak (2006) we have

$$\|\ell(g, \vec{w_2}) - \ell(g, \vec{w_1}) - \langle \nabla_{w_1}\ell(g, \vec{w_1}), \vec{w_2} - \vec{w_1}\rangle - \langle \nabla^2_{w_1}\ell(g, \vec{w_1})(\vec{w_2} - \vec{w_1}), \vec{w_2} - \vec{w_1}\rangle\|$$

$$\leq \frac{\rho}{6}\|\vec{w_2} - \vec{w_1}\|^3 \quad (40)$$

Only considering the upper bound we have:

$$\ell(\vec{w_2}) - \ell(\vec{w_1}) - \langle \nabla_{w_1}\ell(\vec{w_1}), \vec{w_2} - \vec{w_1}\rangle - \langle \nabla^2_{w_1}\ell(\vec{w_1})(\vec{w_2} - \vec{w_1}), \vec{w_2} - \vec{w_1}\rangle \leq \frac{\rho}{6}\|\vec{w_2} - \vec{w_1}\|^3$$

we can rewrite it as:

$$\ell(\vec{w_2}) \leq \frac{\rho}{6}\|\vec{w_2} - \vec{w_1}\|^3 + \ell(\vec{w_1}) + \langle \nabla_{w_1}\ell(\vec{w_1}), \vec{w_2} - \vec{w_1}\rangle + \langle \nabla^2_{w_1}\ell(\vec{w_1})(\vec{w_2} - \vec{w_1}), \vec{w_2} - \vec{w_1}\rangle$$

With SGD we have the update equation given by $\vec{w_2} = \vec{w_1} - \eta\tilde{\nabla}\ell(\vec{w_1})$, using this in the previous step we get:

$$\ell(\vec{w_2}) \leq \frac{\eta^3\rho}{6}\|\tilde{\nabla}_{w_1}\ell(\vec{w_1})\|^3 + \ell(\vec{w_1}) - \eta\langle \nabla_{w_1}\ell(\vec{w_1}), \tilde{\nabla}_{w_1}\ell(\vec{w_1})\rangle + \eta^2\langle \nabla^2_{w_1}\ell(\vec{w_1})\tilde{\nabla}_{w_1}\ell(\vec{w_1}), \tilde{\nabla}_{w_1}\ell(\vec{w_1})\rangle$$

Taking the expectation over $t$, we get:

$$\mathbb{E}_t[\ell(\vec{w_2}) - \ell(\vec{w_1})]$$

$$\leq \frac{\eta^3\rho}{6}\mathbb{E}_t[\|\tilde{\nabla}_{w_1}\ell(w_1)\|^3] - \eta\mathbb{E}_t[\langle \nabla\ell(w_1), \tilde{\nabla}\ell(\vec{w_1})\rangle] + \eta^2\,\mathbb{E}_t[\langle \nabla^2\ell(\vec{w_1})\tilde{\nabla}_{\vec{w_1}}\ell(\vec{w_1}), \tilde{\nabla}_{w_1}\ell(\vec{w_1})\rangle]$$

For ease of writing we split the upper bound into three terms.

**1. First term:** $\mathbb{E}_t[\|\tilde{\nabla}_{w_1}\ell(\vec{w_1})\|^3]$. Given Assumption 18 we have $\mathbb{E}_t[\|\tilde{\nabla}_{w_1}\ell(\vec{w_1})\|^3] \leq \Gamma^3$

**2. Second term:** $\mathbb{E}_t[\langle \nabla_{w_1}\ell(\vec{w_1}), \tilde{\nabla}_{w_1}\ell(\vec{w_1})\rangle]$. Since $\mathbb{E}_t[\tilde{\nabla}_{w_1}\ell(w_1)] = \nabla_{w_1}\ell(\vec{w_1})$, we have

$$\mathbb{E}_t[\langle \nabla_{w_1}\ell(\vec{w_1}), \tilde{\nabla}_{w_1}\ell(\vec{w_1})\rangle] = \|\nabla_{w_1}\ell(\vec{w_1})\|^2$$

**3. Third term:** $\mathbb{E}_t[\langle \nabla^2_{w_1}\ell(\vec{w_1})\tilde{\nabla}_{w_1}\ell(\vec{w_1}), \tilde{\nabla}_{w_1}\ell(\vec{w_1})\rangle]$

Using the formula for the expectation of a quadratic form involving the Hessian:

$$\mathbb{E}[\mathbf{J}^T A \mathbf{J}] = \text{tr}(A\Sigma) + \mu^T A\mu,$$

where $\mathbf{J} = \tilde{\nabla}\ell(\vec{w_1})$, $A = \nabla^2\ell(\vec{w_1})$, $\mu = \nabla\ell(\vec{w_1})$, and $\Sigma$ is the covariance matrix of $\tilde{\nabla}_{w_1}\ell(\vec{w_1})$.

Given $\mathbb{E}_t[\tilde{\nabla}_{w_1}\ell(\vec{w_1})] = \nabla_{w_1}\ell(\vec{w_1})$ and the variance bound $\mathbb{E}_t[\|\tilde{\nabla}_{w_1}\ell(\vec{w_1}) - \nabla_{w_1}\ell(\vec{w_1})\|^2] \leq \sigma^2$, we get:

$$
\begin{aligned}
\mathbb{E}_t[\langle \nabla^2_{w_1}\ell(\vec{w_1})\tilde{\nabla}_{w_1}\ell(\vec{w_1}), \tilde{\nabla}_{w_1}\ell(\vec{w_1})\rangle] &= \text{tr}(\nabla^2_{w_1}\ell(\vec{w_1})\Sigma) + \nabla_{w_1}\ell(\vec{w_1})^T\nabla^2_{w_1}\ell(\vec{w_1})\nabla_{w_1}\ell(\vec{w_1}) \\
&\leq \|\nabla^2_{w_1}\ell(\vec{w_1})\|\text{tr}(\Sigma) + \|\nabla_{w_1}\ell(\vec{w_1})\|^2\|\nabla^2_{w_1}\ell(\vec{w_1})\| \\
&\leq \|\nabla^2_{w_1}\ell(\vec{w_1})\|\sigma^2 + \|\nabla^2_{w_1}\ell(\vec{w_1})\|\|\nabla_{w_1}\ell(\vec{w_1})\|^2 \\
&\leq \|\nabla^2_{w_1}\ell(\vec{w_1})\|(\sigma^2 + \|\nabla_{w_1}\ell(\vec{w_1})\|^2)
\end{aligned}
$$

Substituting these results back into the original inequality:

$$\mathbb{E}_t[\ell(\vec{w_2}) - \ell(\vec{w_1})] \leq \frac{\eta^3\Gamma^3\rho}{6} - \eta\|\nabla_{w_1}\ell(\vec{w_1})\|^2 + \eta^2\|\nabla^2_{w_1}\ell(\vec{w_1})\|(\sigma^2 + \|\nabla_{w_1}\ell(\vec{w_1})\|^2)$$

Grouping the terms

$$\mathbb{E}_t[\ell(\vec{w_2}) - \ell(\vec{w_1})] \leq \frac{\eta^3\Gamma^3\rho}{6} - (\eta - \eta^2\|\nabla^2_{w_1}\ell(\vec{w_1})\|)\|\nabla_{w_1}\ell(\vec{w_1})\|^2 + \eta^2\|\nabla^2_{w_1}\ell(\vec{w_1})\|\sigma^2$$

Let $\Delta\ell_t = \ell(\vec{w}_{t+1}) - \ell(\vec{w}_t)$. Summing this inequality over $T$ iterations, we get:

$$\sum_{t=0}^{T-1}\mathbb{E}_t[\Delta\ell_t] \leq \sum_{t=0}^{T-1}\left(\frac{\eta^3\Gamma^3\rho}{6} - (\eta - \eta^2\|\nabla^2_{w_t}\ell(\vec{w_t})\|)\|\nabla_{w_t}\ell(\vec{w_t})\|^2 + \eta^2\|\nabla^2_{w_t}\ell(\vec{w_t})\|\sigma^2\right)$$

Telescoping the sum, the left-hand side telescopes:

$$\mathbb{E}[\ell(\vec{w}_T) - \ell(\vec{w}_0)] \le \sum_{t=0}^{T-1} \left( \frac{\eta^3 \Gamma^3 \rho}{6} - (\eta - \eta^2 \|\nabla_{w_t}^2 \ell(\vec{w}_t)\|) \|\nabla_{w_t} \ell(\vec{w}_t)\|^2 + \eta^2 \|\nabla_{w_t}^2 \ell(\vec{w}_t)\|\sigma^2 \right)$$

$$\mathbb{E}[\ell(\vec{w}_T) - \ell(\vec{w}_0)] \le \frac{\eta^3 \Gamma^3 \rho T}{6} + \eta^2 \sum_{t=0}^{T-1} \|\nabla_{w_t}^2 \ell(\vec{w}_t)\| \|\nabla_{w_t} \ell(\vec{w}_t)\|^2 - \eta \sum_{t=0}^{T-1} \|\nabla_{w_t} \ell(\vec{w}_t)\|^2$$

$$+ \eta^2 \sigma^2 \sum_{t=0}^{T-1} \|\nabla_{w_t}^2 \ell(\vec{w}_t)\|$$

$$\mathbb{E}[\ell(\vec{w}_T) - \ell(\vec{w}_0)] \le \frac{\eta^3 \Gamma^3 \rho T}{6} - \eta \sum_{t=0}^{T-1} \|\nabla_{w_t} \ell(\vec{w}_t)\|^2 + \eta^2(\sigma^2 + \Gamma^2) \sum_{t=0}^{T-1} \|\nabla_{w_t}^2 \ell(\vec{w}_t)\|$$

$$\eta \sum_{t=0}^{T-1} \|\nabla_{w_t} \ell(\vec{w}_t)\|^2 \le \frac{\eta^3 \Gamma^3 \rho T}{6} + \mathbb{E}[\ell(\vec{w}_0) - \ell(\vec{w}_T)] + \eta^2(\sigma^2 + \Gamma^2) \sum_{t=0}^{T-1} \|\nabla_{w_t}^2 \ell(\vec{w}_t)\|$$

$$\frac{\mathcal{M} T_{max} k_g}{2\lambda \ell(\vec{w}_0)} \sum_{t=0}^{T-1} \|\nabla_{w_t} \ell(\vec{w}_t)\|^2 \le \frac{\eta^2 k_g \Gamma^3 \rho T \mathcal{M} T_{max}}{12 \lambda \ell(\vec{w}_0)} + \frac{\mathcal{M} T_{max} k_g}{2\lambda \eta \ell(\vec{w}_0)} \mathbb{E}[\ell(\vec{w}_0) - \ell(\vec{w}_T)]$$

$$+ \frac{\eta k_g (\sigma^2 + \Gamma^2) \mathcal{M} T_{max}}{2\lambda \ell(\vec{w}_0)} \sum_{t=0}^{T-1} \|\nabla_{w_t}^2 \ell(\vec{w}_t)\|$$

$$\frac{\mathcal{M} T_{max} k_g}{2\lambda \ell(\vec{w}_0)} \sum_{t=0}^{T-1} \|\nabla_{w_t} \ell(\vec{w}_t)\|^2 \le C_1' + C_2' \mathbb{E}\left[1 - \frac{\ell(\vec{w}_T)}{\ell(\vec{w}_0)}\right] + C_3' \sum_{t=0}^{T-1} \|\nabla_{w_t}^2 \ell(\vec{w}_t)\|$$

$$\frac{\mathcal{M} T_{max}}{2\lambda \ell(\vec{w}_0)} \sum_{t=0}^{T-1} \|\nabla_X \ell(\vec{w}_t)\|^2 \le C_1' + C_2' + C_3' \sum_{t=0}^{T-1} \|\nabla_{w_t}^2 \ell(\vec{w}_t)\|$$

$$\frac{\mathcal{M} T_{max}}{2\lambda \ell(\vec{w}_0)} \sum_{t=0}^{T-1} \|\nabla_X \ell(\vec{w}_t)\|^2 \le C_1 + C_3' \sum_{t=0}^{T-1} \|\nabla_{w_t}^2 \ell(\vec{w}_t)\|$$

$$\frac{\mathcal{M} T_{max}}{2\lambda \ell(\vec{w}_0)} \sum_{t=0}^{T-1} \|\nabla_X \ell(\vec{w}_t)\|^2 \le C_1 + C_3' k_h \sum_{t=0}^{T-1} \frac{1}{k_h} \|\nabla_{w_t}^2 \ell(\vec{w}_t)\|$$

$$\frac{\mathcal{M} T_{max}}{2\lambda \ell(\vec{w}_0)} \sum_{t=0}^{T-1} \|\nabla_X \ell(\vec{w}_t)\|^2 \le C_1 + C_2 \sum_{t=0}^{T-1} \|\nabla_X^2 \ell(\vec{w}_t)\|$$

Using the result from Theorem 4.3 we have

$$T_a \le \frac{\mathcal{M} T_{max}}{2\lambda \ell(\vec{w}_0)} \sum_{t=0}^{T_a - 1} \|\nabla_X \ell(\vec{w}_t)\|^2 \le C_1 + C_2 \sum_{t=0}^{T_a - 1} \|\nabla_X^2 \ell(\vec{w}_t)\| \quad \blacksquare$$

### E.6   MEMORIZATION AND LOSS GRADIENT

**Proof of Theorem 4.5**

**Assumptions:**

- Assume that the loss function $\ell$ is $\mathcal{L}$-Lipschitz.
- Assume error stability as stated in Assumption 1.
- Assume uniform model bias as stated in Assumption 15.
- Assume generalization as stated in Assumption 14.
- Assume two samples are within an $\alpha$ ball of each other.
- Assume loss is bounded $0 \le \ell \le L$

**Proof**

From Lemma A.2 from Ravikumar et al. (2024a) we have

$$\mathbb{E}_\phi[\ell(g_S^\phi, \vec{z_i}) - \ell(g_{S^{\setminus i}}^\phi, \vec{z_j})] \leq m\beta + (4m-1)\gamma + 2(m-1)\Delta \tag{41}$$

Using $\mathcal{L}$-lipschitzness of the loss function we have

$$-\frac{\mathcal{L}}{2}\|\vec{z_1} - \vec{z_2}\|^2 \leq \ell(g, \vec{z_1}) - \ell(g, \vec{z_2}) - \langle \nabla_{z_2}\ell(g, \vec{z_2}), \vec{z_1} - \vec{z_2}\rangle \leq \frac{\mathcal{L}}{2}\|\vec{z_1} - \vec{z_2}\|^2$$

$$\ell(g, \vec{z_1}) \leq \frac{\mathcal{L}}{2}\|\vec{z_1} - \vec{z_2}\|^2 + \langle \nabla_{z_2}\ell(g, \vec{z_2}), \vec{z_1} - \vec{z_2}\rangle + \ell(g, \vec{z_2})$$

$$\ell(g, \vec{z_1}) \leq \frac{\mathcal{L}}{2}\|\alpha\|^2 + \|\alpha\|\|\nabla_{z_2}\ell(g, \vec{z_2})\| + \ell(g, \vec{z_2})$$

$$\ell(g, \vec{z_j}) \leq \frac{\mathcal{L}}{2}\|\alpha\|^2 + \|\alpha\|\|\nabla_{z_i}\ell(g, \vec{z_i})\| + \ell(g, \vec{z_i})$$

Using this result in Equation 41 we have:

$$\mathbb{E}_\phi[\ell(g_S^\phi, \vec{z_i}) - \ell(g_{S^{\setminus i}}^\phi, \vec{z_j})] \leq m\beta + (4m-1)\gamma + 2(m-1)\Delta$$

$$\mathbb{E}_\phi\left[\ell(g_S^\phi, \vec{z_i}) - \ell(g_{S^{\setminus i}}^\phi, \vec{z_i}) - \frac{\mathcal{L}}{2}\|\alpha\|^2 - \|\alpha\|\|\nabla_{z_i}\ell(g_{S^{\setminus i}}^\phi, \vec{z_i})\|\right] \leq \mathbb{E}_\phi[\ell(g_S^\phi, \vec{z_i}) - \ell(g_{S^{\setminus i}}^\phi, \vec{z_j})]$$

$$\mathbb{E}_\phi\left[\ell(g_S^\phi, \vec{z_i}) - \ell(g_{S^{\setminus i}}^\phi, \vec{z_i}) - \frac{\mathcal{L}}{2}\|\alpha\|^2 - \|\alpha\|\|\nabla_{z_i}\ell(g_{S^{\setminus i}}^\phi, \vec{z_i})\|\right] \leq m\beta + (4m-1)\gamma + 2(m-1)\Delta$$

$$\mathbb{E}_\phi\left[\ell(g_S^\phi, \vec{z_i}) - \ell(g_{S^{\setminus i}}^\phi, \vec{z_i})\right] \leq m\beta + (4m-1)\gamma + 2(m-1)\Delta$$

$$+ \frac{\mathcal{L}}{2}\|\alpha\|^2 + \|\alpha\|\,\mathbb{E}_\phi\left[\|\nabla_{z_i}\ell(g_{S^{\setminus i}}^\phi, \vec{z_i})\|\right]$$

Thus using the result from Ravikumar et al. (2024a) be we

$$L\,\mathrm{mem}(S, \vec{z_i}) \leq m\beta + (4m-1)\gamma + 2(m-1)\Delta + \frac{\mathcal{L}}{2}\|\alpha\|^2 + \|\alpha\|\,\mathbb{E}_\phi\left[\|\nabla_{z_i}\ell(g_{S^{\setminus i}}^\phi, \vec{z_i})\|\right]$$

$$\mathrm{mem}(S, \vec{z_i}) \leq \frac{m\beta}{L} + \frac{(4m-1)\gamma}{L} + \frac{2(m-1)\Delta}{L} + \frac{\mathcal{L}}{2L}\|\alpha\|^2 + \frac{\|\alpha\|}{L}\mathbb{E}_\phi\left[\|\nabla_{z_i}\ell(g_{S^{\setminus i}}^\phi, \vec{z_i})\|\right]$$

$$\mathrm{mem}(S, \vec{z_i}) \leq C_3 + C_4\,\mathbb{E}_\phi\left[\|\nabla_{z_i}\ell(g_{S^{\setminus i}}^\phi, \vec{z_i})\|\right] \quad \blacksquare$$

E.7    MEMORIZATION AND LOSS

**Proof of Theorem 4.4**

**Assumptions:**

- Assume that the loss function $\ell$ is $\mathcal{L}$-Lipschitz.
- Assume error stability as stated in Assumption 1.
- Assume that the gradient estimator $\widetilde{\nabla}_w\ell$ is unbiased i.e., $\mathbb{E}_t[\widetilde{\nabla}_w\ell(\vec{w_t})] = \nabla_w\ell(\vec{w_t})$.
- Assume bounded gradient as stated in Assumption 18.
- Assume loss is bounded $0 \leq \ell \leq L$

**Proof**

Consider Equation 33

$$T_{z_a}\,\mathbb{E}_t\left[\ell(\vec{w_0}) - \ell(\vec{w}_{T_{z_b}})\right] = T_{z_b}\,\mathbb{E}_t\left[\ell(\vec{w_0}) - \ell(\vec{w}_{T_a})\right]$$

$$\mathbb{E}_\phi\left[\ell(\vec{w_0})(T_{z_a} - T_{z_b})\right] = -\hat{T}_{z_a}L\,\mathrm{mem}(S, \vec{z_b}) + \hat{T}_{z_b}L\,\mathrm{mem}(S, \vec{z_a})$$

$$+ \mathbb{E}_\phi\left[T_{z_b}\ell^{\setminus a}(\vec{w}_{S^{\setminus a}}^*) - T_{z_a}\ell^{\setminus b}(\vec{w}_{S^{\setminus b}}^*)\right]$$

where $\ell^{\backslash i}(\vec{w}_t) = \mathbb{E}_{z \in S^{\backslash i}}[\ell(\vec{w}_t, \vec{z})]$ denotes the loss function when the $\vec{z}_i^{th}$ sample was removed from the dataset. Let $\vec{w}^*$ denote such a model at convergence. Then we have

$$\hat{T}_{z_a} \mathbb{E}_\phi [\ell(\vec{w}_0)] - \hat{T}_{z_b} \left( \mathbb{E}_\phi [\ell(\vec{w}_0)] + \mathbb{E}_\phi \left[ \ell^{\backslash a}(\vec{w}_{S\backslash a}^*) \right] \right) = -\hat{T}_{z_a} L \operatorname{mem}(S, \vec{z}_b) + \hat{T}_{z_b} L \operatorname{mem}(S, \vec{z}_a)$$
$$- \hat{T}_a \mathbb{E}_\phi \left[ \ell^{\backslash b}(\vec{w}_{S\backslash b}^*) \right] \tag{42}$$

Where $\hat{T}_z = \mathbb{E}_\phi[T_z]$, i.e. expected learning time of a sample. We get a lower bound on $\hat{T}_{z_b} \left( \mathbb{E}_\phi [\ell(\vec{w}_0)] + \mathbb{E}_\phi \left[ \ell^{\backslash a}(\vec{w}^*) \right] \right)$ using our stability relation

$$\hat{T}_{z_b} \left( \mathbb{E}_\phi [\ell(\vec{w}_0)] + \mathbb{E}_\phi \left[ \ell^{\backslash a}(\vec{w}_{S\backslash a}^*) \right] \right) \geq \hat{T}_{z_b} \left( \mathbb{E}_\phi [\ell(\vec{w}_0)] - \beta + \mathbb{E}_\phi [\ell(\vec{w}_S^*)] \right)$$

Note $\vec{w}_S^* \neq \vec{w}_{S\backslash a}^*$. This is because $\vec{w}_S^*, \vec{w}^*$ are the result of optimizing on different datasets $S$ and $S^{\backslash a}$ respectively. Thus we have

$$-\hat{T}_{z_a} L \operatorname{mem}(S, \vec{z}_b) + \hat{T}_{z_b} L \operatorname{mem}(S, \vec{z}_a) - \hat{T}_{z_a} \mathbb{E}_\phi \left[ \ell^{\backslash b}(\vec{w}_{S\backslash b}^*) \right] \leq \hat{T}_{z_a} \mathbb{E}_\phi [\ell(\vec{w}_0)]$$
$$- \hat{T}_{z_b} \left( \mathbb{E}_\phi [\ell(\vec{w}_0)] - \beta + \mathbb{E}_\phi [\ell(\vec{w}_S^*)] \right)$$

$$\hat{T}_{z_b} L \operatorname{mem}(S, \vec{z}_a) - \hat{T}_a \mathbb{E}_\phi \left[ \ell^{\backslash b}(\vec{w}_{S\backslash b}^*) \right] \leq \hat{T}_{z_a} \mathbb{E}_\phi [\ell(\vec{w}_0)] - \hat{T}_{z_b} \left( \mathbb{E}_\phi [\ell(\vec{w}_0)] - \beta + \mathbb{E}_\phi [\ell(\vec{w}_S^*)] \right)$$
$$+ \hat{T}_{z_a} L \operatorname{mem}(S, \vec{z}_b)$$
$$\hat{T}_{z_b} L \operatorname{mem}(S, \vec{z}_a) \leq \hat{T}_{z_a} \mathbb{E}_\phi \left[ \ell^{\backslash b}(\vec{w}_{S\backslash b}^*) \right] + \hat{T}_{z_a} \mathbb{E}_\phi [\ell(\vec{w}_0)] - \hat{T}_{z_b} \left( \mathbb{E}_\phi [\ell(\vec{w}_0)] \right)$$
$$- \hat{T}_{z_b} \beta + \hat{T}_{z_b} \mathbb{E}_\phi [\ell(\vec{w}_S^*)] + \hat{T}_{z_a} L \operatorname{mem}(S, \vec{z}_b)$$

Since the loss is bound and $\operatorname{mem} \leq 1$ we have

$$\hat{T}_{z_b} L \operatorname{mem}(S, \vec{z}_a) \leq \hat{T}_{z_a} L + \hat{T}_{z_a} L - \hat{T}_{z_b} \left( \mathbb{E}_\phi [\ell(\vec{w}_0)] \right) - \hat{T}_{z_b} \beta + \hat{T}_{z_b} \mathbb{E}_\phi [\ell(\vec{w}_S^*)] + \hat{T}_{z_a} L$$
$$\hat{T}_{z_b} L \operatorname{mem}(S, \vec{z}_a) \leq 3\hat{T}_{z_a} L - \hat{T}_{z_b} \left( \mathbb{E}_\phi [\ell(\vec{w}_0)] \right) - \hat{T}_{z_b} \beta + \hat{T}_{z_b} \mathbb{E}_\phi [\ell(\vec{w}_S^*)]$$
$$\leq 3\hat{T}_{z_a} L - \hat{T}_{z_b} \beta + \hat{T}_{z_b} \mathbb{E}_\phi [\ell(\vec{w}_S^*) - \ell(\vec{w}_0)]$$
$$\leq 3\hat{T}_{z_a} L - \hat{T}_{z_b} \beta + \hat{T}_{z_b} \sum_{t=0}^{*} \mathbb{E}_\phi [\ell(\vec{w}_{t+1}) - \ell(\vec{w}_t)]$$
$$\leq 3\hat{T}_{z_a} L - \hat{T}_{z_b} \beta + \hat{T}_{z_b} \sum_{t=0}^{*} \mathbb{E}_\phi [\ell(\vec{w}_{t+1})] - \hat{T}_{z_b} \sum_{t=0}^{*} \mathbb{E}_\phi [\ell(\vec{w}_t)]$$

If the loss estimation using a single sample has a variance $\sigma_l$, then using the Chebyshev inequality with a confidence of $1 - \delta$

$$\hat{T}_{z_b} L \operatorname{mem}(S, \vec{z}_a) \leq 3\hat{T}_{z_a} L - \hat{T}_{z_b} \beta + \hat{T}_{z_b} \sum_{t=0}^{*} \mathbb{E}_\phi [\ell(\vec{w}_{t+1})] - \hat{T}_{z_b} \sum_{t=0}^{*} \left( \mathbb{E}_\phi [\ell(\vec{w}_t, \vec{z}_a)] - \frac{\sigma_l}{\sqrt{\delta}} \right)$$
$$\leq 3\hat{T}_{z_a} L - \hat{T}_{z_b} \beta + \hat{T}_{z_b} \sum_{t=0}^{*} \mathbb{E}_\phi [\ell(\vec{w}_{t+1})] - \hat{T}_{z_b} \operatorname{CSL}(\vec{z}_a) + \hat{T}_{z_b} \frac{\sigma_l}{\sqrt{\delta}}$$

From Theorem 4.1, $\hat{T}_{z_a}$ depends on memorization given by

$$\hat{T}_{z_a} \leq \frac{\hat{T}_{ref} L}{\mathbb{E}_\phi [\ell(\vec{w}_0)]} \left( \operatorname{mem}(S, \vec{z}_a) + 1 + \frac{2\sigma_l}{L\sqrt{\delta}} \right) + \hat{T}_{ref}$$

$$\hat{T}_{z_b} L \operatorname{mem}(S, \vec{z}_a) \le 3L \left( \frac{\hat{T}_{ref} L}{\mathbb{E}_\phi \left[ \ell(\vec{w}_0) \right]} \left( \operatorname{mem}(S, \vec{z}_a) + 1 + \frac{2\sigma_l}{L\sqrt{\delta}} \right) + \hat{T}_{ref} \right) - \hat{T}_{z_b} \beta$$

$$+ \hat{T}_{z_b} \sum_{t=0}^{*} \mathbb{E}_\phi \left[ \ell(\vec{w}_{t+1}) \right] - \hat{T}_{z_b} \operatorname{CSL}(\vec{z}_a) + \hat{T}_{z_b} \frac{\sigma_l}{\sqrt{\delta}}$$

$$\hat{T}_{z_b} L \operatorname{mem}(S, \vec{z}_a) \le \frac{3L^2 \hat{T}_{ref}}{\mathbb{E}_\phi \left[ \ell(\vec{w}_0) \right]} \operatorname{mem}(S, \vec{z}_a) + \frac{3L^2 \hat{T}_{ref}}{\mathbb{E}_\phi \left[ \ell(\vec{w}_0) \right]} + 3L\hat{T}_{ref} - \hat{T}_{z_b} \beta$$

$$+ \hat{T}_{z_b} \sum_{t=0}^{*} \mathbb{E}_\phi \left[ \ell(\vec{w}_{t+1}) \right] - \hat{T}_{z_b} \operatorname{CSL}(\vec{z}_a) + \hat{T}_{z_b} \frac{\sigma_l}{\sqrt{\delta}} + \frac{6\hat{T}_{ref} L \sigma_l}{\mathbb{E}_\phi [\ell(\vec{w}_0)] \sqrt{\delta}}$$

$$\hat{T}_{z_b} \operatorname{CSL}(\vec{z}_a) - \hat{T}_{z_b} \frac{\sigma_l}{\sqrt{\delta}} \le \left( \frac{3L^2 \hat{T}_{ref}}{\mathbb{E}_\phi \left[ \ell(\vec{w}_0) \right]} - \hat{T}_{z_b} L \right) \operatorname{mem}(S, \vec{z}_a) + \frac{3L^2 \hat{T}_{ref}}{\mathbb{E}_\phi \left[ \ell(\vec{w}_0) \right]} + 3L\hat{T}_{ref}$$

$$- \hat{T}_{z_b} \beta + \hat{T}_{z_b} \sum_{t=0}^{*} \mathbb{E}_\phi \left[ \ell(\vec{w}_{t+1}) \right] + \frac{6\hat{T}_{ref} L \sigma_l}{\mathbb{E}_\phi [\ell(\vec{w}_0)] \sqrt{\delta}}$$

$$\hat{T}_{z_b} \operatorname{CSL}(\vec{z}_a) - \left( \hat{T}_{z_b} + \frac{6\hat{T}_{ref} L}{\mathbb{E}_\phi [\ell(\vec{w}_0)]} \right) \frac{\sigma_l}{\sqrt{\delta}} - \frac{3L^2 \hat{T}_{ref}}{\mathbb{E}_\phi \left[ \ell(\vec{w}_0) \right]} - 3L\hat{T}_{ref} + \hat{T}_{z_b} \beta - \hat{T}_{z_b} \sum_{t=0}^{*} \mathbb{E}_\phi \left[ \ell(\vec{w}_{t+1}) \right] \le$$

$$\left( \frac{3L^2 \hat{T}_{ref}}{\mathbb{E}_\phi \left[ \ell(\vec{w}_0) \right]} - \hat{T}_{z_b} L \right) \operatorname{mem}(S, \vec{z}_a)$$

Let $\hat{T}_{z_b} = \hat{T}_{ref}$. Thus we have

$$\hat{T}_{ref} \operatorname{CSL}(\vec{z}_a) - \left( \hat{T}_{ref} + \frac{6\hat{T}_{ref} L}{\mathbb{E}_\phi [\ell(\vec{w}_0)]} \right) \frac{\sigma_l}{\sqrt{\delta}} - \frac{3L^2 \hat{T}_{ref}}{\mathbb{E}_\phi \left[ \ell(\vec{w}_0) \right]} - 3L\hat{T}_{ref} + \hat{T}_{ref} \beta - \hat{T}_{ref}^2 \ell(\vec{w}^*) \le$$

$$\left( \frac{3L^2 \hat{T}_{ref}}{\mathbb{E}_\phi \left[ \ell(\vec{w}_0) \right]} - \hat{T}_{ref} L \right) \operatorname{mem}(S, \vec{z}_a)$$

This result comes from the lower bound established on the loss at each iteration. Now divide each size by $\hat{T}_{ref}$

$$\operatorname{CSL}(\vec{z}_a) - \left( 1 + \frac{6L}{\mathbb{E}_\phi \left[ \ell(\vec{w}_0) \right]} \right) \frac{\sigma_l}{\sqrt{\delta}} - \frac{3L^2}{\mathbb{E}_\phi \left[ \ell(\vec{w}_0) \right]} - 3L + \beta - \hat{T}_{ref} \ell(\vec{w}^*) L \le$$

$$\left( \frac{3L^2}{\mathbb{E}_\phi \left[ \ell(\vec{w}_0) \right]} - L \right) \operatorname{mem}(S, \vec{z}_a)$$

This we have

$$C_5 \left( \operatorname{CSL}(\vec{z}_a) - C_6 + C_7 \right) \le \operatorname{mem}(S, \vec{z}_a) \quad \blacksquare$$

Where $C_6$ is the lower bound on the loss at each step, thus the term $\operatorname{CSL} - C_6$ estimates how far each step's loss is from the lower bound. And $C_7$ is the total estimation error. $C_5$ is a scaling term to bring the result in the correct range.

$$C_5 = \frac{1}{\left( \dfrac{3L^2}{\mathbb{E}_\phi \left[ \ell(\vec{w}_0) \right]} - L \right)} \tag{43}$$

$$C_6 = \hat{T}_{ref} \ell(\vec{w}^*) \tag{44}$$

$$C_7 = \left( \beta - \frac{3L^2}{\mathbb{E}_\phi \left[ \ell(\vec{w}_0) \right]} - 3L - \left( 1 + \frac{6L}{\mathbb{E}_\phi \left[ \ell(\vec{w}_0) \right]} \right) \frac{\sigma_l}{\sqrt{\delta}} \right) \tag{45}$$

