# OpenReview forum: "Memorization and the Orders of Loss: A Learning Dynamics Perspective"
_ICLR.cc/2025/Conference — Submitted to ICLR 2025_

### Official Review · Reviewer_uKr7 · 2024-11-04

**Soundness:** 2
**Presentation:** 2
**Contribution:** 2
**Rating:** 3
**Confidence:** 2

**Summary:**

This paper presents a novel approach to analyzing memorization in deep learning models by introducing Cumulative Sample Loss (CSL) and Cumulative Sample Gradient (CSG) as proxies. The authors theoretically and empirically demonstrate the relationship of these proxies to stability-based memorization and learning time, showing that CSL and CSG are computationally efficient alternatives. The framework is validated on a range of computer vision tasks, including the detection of mislabeled and duplicate samples, underscoring its potential for practical dataset curation.

**Strengths:**

**S1.** The submission tackles the important problem of memorization in deep learning, which has practical and theoretical relevance, particularly for privacy and interpretability.

**S2.** The proposed CSL and CSG metrics significantly reduce computational demands compared to previous stability-based memorization metrics.

**S3.** The experimental evaluation is diverse, demonstrating that CSL and CSG can effectively identify mislabeled and duplicate samples, which is useful for curating and refining datasets.

**Weaknesses:**

**W1.** The theoretical claims in this paper depend on several restrictive assumptions—including bounded loss, bounded expected stochastic gradient moments, and stability—that are challenging to ensure in practice and are particularly questionable in deep learning context. For example, a quadratic loss (or MSE) for a linear model does not satisfy these assumptions, as it is unbounded. Additionally, Theorem 4.6 relies on the $\mu$-PL condition, a slight relaxation of strong convexity, which, when combined with bounded gradient norm assumptions, results in an basically empty class of functions.

**W2.** The paper’s mathematical notation is dense and occasionally ambiguous, which may limit accessibility and readability. Key terminology (e.g., “loss estimation variance,” “expected learning time,” $\vec{w}^*$, and $\alpha$-Ball) is introduced without adequate definition, and the dependence of bounds on various constants makes the theorems challenging to follow. Additionally, some of these constants can be arbitrarily large, rendering certain theoretical results less insightful.

The proposed sample learning condition (4) is very questionable to me as the stochastic gradient norm can be arbitrary large even at the optimum for convex models.

**W3.** Some claims regarding the linear relationships between the metrics are not adequately justified, as the theoretical results mostly provide upper bounds in worst-case scenarios rather than precise relationships. The statement that loss
> Loss serves as **the most compute-efficient** proxy for measuring memorization

seems overly strong and requires refinement for accuracy.

#### Minor comments
- Consider providing a clearer explanation of the memorization “metric,” as it is challenging to understand on first reading.

**Questions:**

1. Why is the bound on the stochastic gradient norm (Equation 18) defined for the third power? This is unconventional.

2. Does the SGD optimization apply to a finite sum (empirical risk minimization) or a stochastic problem?

---

> ### Author Response · Authors · 2024-11-23
>
> We thank the reviewer for their feedback. We address the weaknesses and questions below:
>
> W1. The theoretical claims in this paper depend on several restrictive assumptions—including bounded loss, bounded expected stochastic gradient moments, and stability—that are challenging to ensure in practice and are particularly questionable in deep learning context. For example, a quadratic loss (or MSE) for a linear model does not satisfy these assumptions, as it is unbounded. Additionally, Theorem 4.6 relies on the $mu$-PL condition, a slight relaxation of strong convexity, which, when combined with bounded gradient norm assumptions, results in an basically empty class of functions.
>
> **A:** We would like to clarify that the $\mu$-PL condition is not assumed for all theorems but specifically applies to Theorems 4.3 and 4.6. Additionally, there are loss functions that satisfy the combined set of assumptions, such as the 0-1 loss and others, which remain valid within the deep learning framework. For further details, please refer to our discussion in the section “Remark on Assumptions” (Section 4.2.2) where we discuss the validity of our assumptions in a deep learning setting.
>
> W2. The paper’s mathematical notation is dense and occasionally ambiguous, which may limit accessibility and readability. Key terminology (e.g., “loss estimation variance,” “expected learning time,”
> , and $\alpha$-Ball) is introduced without adequate definition, and the dependence of bounds on various constants makes the theorems challenging to follow. Additionally, some of these constants can be arbitrarily large, rendering certain theoretical results less insightful.
> The proposed sample learning condition (4) is very questionable to me as the stochastic gradient norm can be arbitrary large even at the optimum for convex models.
>
> **A:** Thank you for your feedback. The definitions of the additional terms are provided in the appendix under the "Additional Background" section, as space constraints prevented including them in the main text. Regarding the learning condition, please note that it refers to the average gradient norm per iteration (i.e., the sum divided by $T$), not the cumulative sum of gradient norms. As shown in Theorem B.1, this converges as $T \to \infty$ for SGD under non-convex settings. This makes it a suitable definition for the learning condition.
>
> W3. Some claims regarding the linear relationships between the metrics are not adequately justified, as the theoretical results mostly provide upper bounds in worst-case scenarios rather than precise relationships. The statement that loss
> "Loss serves as the most compute-efficient proxy for measuring memorization"
> seems overly strong and requires refinement for accuracy.
>
> **A:** Thank you for your feedback we have revised it to state “Loss serves as the most compute-efficient proxy among the proxies considered”.
>
> Questions:
>
>  1.	Why is the bound on the stochastic gradient norm (Equation 18) defined for the third power? This is unconventional.\
>      **A:**  This is needed to be able to link the results of CSL and CSG with input loss curvature [1, 2]. While this assumption is non-traditional for optimization we used this to link this to prior proxies on memorization. We would like to add that while we make use of tools from optimization, our results deal with the memorization aspect of deep learning which is a different and novel application and perspective obtained using the tools of optimization.
>
> 2. Does the SGD optimization apply to a finite sum (empirical risk minimization) or a stochastic problem?
>
>     **A:** In this paper we consider the finite sum problem.
>
>
> [1] Ravikumar et al. "Unveiling Privacy, Memorization, and Input Curvature Links." In Forty-first International Conference on Machine Learning 2024.
>
> [2] Garg et al. "Memorization Through the Lens of Curvature of Loss Function Around Samples." In Forty-first International Conference on Machine Learning 2024.

---

> > ### Comment · Reviewer_uKr7 · 2024-11-28
> >
> > I appreciate the authors response. However my major concerns about the restrictive assumptions were not resolved as I do not find satisfying the discussion “Remark on Assumptions” (Section 4.2.2). Thus, I tend to maintain my score for now.

---

### Official Review · Reviewer_eheR · 2024-11-05

**Soundness:** 1
**Presentation:** 3
**Contribution:** 2
**Rating:** 3
**Confidence:** 4

**Summary:**

The paper proposes two new ways to detect memorization (defined by the leave-one-out score definition proposed in Feldman 2020), which is cumulative loss and cumulative gradient norm of a data point. Theoretical justifications are provided and the empirical results look promising.

**Strengths:**

Trending topic. Memorization detection is important, but it is often inefficient to compute the memorization score defined by the counterfactual leave-one-out score. The paper proposes lightweight method that seems to strongly correlated with the memorization score.

The paper is well-organized and very pleasant to read. Theorems are always followed by proof sketch and implications.

**Weaknesses:**

While the theoretical justifications are provided, I am not sure how meaningful are they. For instance, for Theorem 4.1, the bound includes term $T_{max} L/E[\ell(w_0)] >= T_max$, but isn't $T_{z_i} \le T_{max}$ as that's the max model training iterations? I also don't understand where the T_{ref} comes from.

Some notations are weird. Does $\nabla_X$ means the gradient over input feature or parameter?

While the theorems states a bunch of inequalities between different quantities, I am not sure whether those inequalities can lead to the conclusion that these quantities are linearly correlated with each other.

**Questions:**

See weaknesses.

---

> ### Author Response · Authors · 2024-11-23
>
> We thank the reviewer for their feedback. We address the weaknesses and questions below:
>
> 1. While the theoretical justifications are provided, I am not sure how meaningful are they. For instance, for Theorem 4.1, the bound includes term , but isn't as that's the max model training iterations? I also don't understand where the T_{ref} comes from.
>
>     **A:** We appreciate the reviewer’s observation regarding Theorem 4.1 and agree that the original bound was not tight. We have updated the proof to replace the term 1 with $\frac{2\beta}{L}$, which represents the stability factor. This change ensures that the term inside the brackets is likely less than 1, providing a more meaningful and interpretable result. This has been updated in the revised version.
>
>     As for $T_{\text{ref}}$, it arises because we compare the learning time of two samples. When analyzing the learning time for sample $a$, we express it relative to sample $b$. Since $b$ can be any sample, we select a reference sample from the dataset to measure learning times. This reference sample $T_{\text{ref}}$, can be chosen such that its learning time is very close to 0 (i.e. a very easy training data point for the model).
>
> 2. Some notations are weird. Does $\nabla_X$ means the gradient over input feature or parameter?
>
>     **A:** This refers to the gradient in the input space. We will make sure to clarify this distinction in the revised version for better readability and understanding.
>
> 3. While the theorems states a bunch of inequalities between different quantities, I am not sure whether those inequalities can lead to the conclusion that these quantities are linearly correlated with each other.
>
>     **A:** The theorems provide a series of inequalities between various quantities. While these inequalities don't immediately imply linear correlation, the bounds they establish are linear. If these bounds are tight, then the quantities would indeed exhibit linear correlation. This is precisely what we test in our experiments. Our results strongly suggest that the bounds are likely tight, supporting the hypothesis of linear correlation.

---

> > ### Comment · Reviewer_eheR · 2024-11-24
> >
> > Thanks so much for the response. I tried to verify Theorem 4.1 but still have a hard time. Theorem 4.1 is based on Theorem B.1, which further depends on Lemma D.1. I tried to read its proof, but it's very poorly written. The definition of $s_X$ in the theorem statement is not clear and I only found it in the proof. Even just the first line of the proof, I don't know what it mean by $\nabla_H$. The authors should make sure all technical proofs are clean and clear.

---

### Official Review · Reviewer_Z2Ei · 2024-11-06

**Soundness:** 3
**Presentation:** 4
**Contribution:** 2
**Rating:** 3
**Confidence:** 4

**Summary:**

This paper investigates memorization in deep neural networks through the lens of learning dynamics. The authors propose two computationally efficient metrics - Cumulative Sample Loss (CSL) and Cumulative Sample Gradient (CSG) - to measure memorization effects during training. They establish theoretical bounds connecting these metrics to learning time and stability-based memorization. Through experiments on CIFAR-100 and ImageNet, they demonstrate these metrics can effectively detect memorized examples and are computationally efficient. The practical applications include detecting mislabeled examples and duplicates in datasets, where their approach achieves state-of-the-art performance while being 4 orders of magnitude faster than previous methods.

**Strengths:**

Strengths:
1. Clear, well-defined problem and the approach is novel from what I. know.
2. The writing is clear and the problem is well motivated
3. This is a significant problem to solve since there are multiple approaches to measuring memorization but none that works fast and consistently.

**Weaknesses:**

Weaknesses:
1. My main concern with the paper lies with the empirical evaluation strategy. The authors validate their proposed metrics (CSL/CSG) against Feldman's [1] subsampled influence scores rather than true leave-one-out training. This is problematic because recent work by Basu et al[2] and  Bae et al.[3] has shown that influence functions do not correlate well with actual leave-one-out training for large neural networks. This approximation gets worse for larger networks. Without validation against true leave-one-out training, we cannot verify if their metric actually captures memorization or merely correlates with Feldman's influence scores. Essentially, it would show correlation between two approximations rather than validation against ground truth.
2. The theoretical framework assumes that SGD is trained to convergence which isn't really done in most practical settings. On the theoretical side, on its own, this is fine to prove some theorems (with the caveat that they are idealized). But related to the above point, Bae et al. have shown that exactly this property (among others) causes influence functions to not correlate with leave one out retraining. So there is a gap here between the theoretical properties being proven and the practical aspects. In my opinion, such a gap is okay if there is a novel technical contribution on the theoretical side but most proofs seem to follow: Start with SGD convergence assumptions, Use telescoping sums to relate changes across iterations, Apply basic inequalities (Lipschitz, bounded gradients etc.), Arrive at bounds that relate different quantities. Maybe I'm missing something?
3. On the theoretical side, there is an $\alpha$-adjacency assumption used in the proofs:

  3.1. Requires existence of data points within $\alpha$-balls of each other

  3.2. Authors claim this holds as $\alpha$ is unrestricted, but this makes the assumption essentially meaningless - if $\alpha$ can be arbitrarily large, what does the bound tell us? Ideally, I would have like some empirical validation that real datasets satisfy this property for meaningful $\alpha$ values as well.

4. The authors use arbitrary thresholds (0.25 for memorization, 0.15 for influence) across all datasets.  I did not see a justification for why these specific values were chosen or an analysis of how results change with different thresholds.
5. No error bars on key results, also. only a single architecture (resNet-18) is used in all experiments.
6. The $\mu$-PL condition is not defined anywhere (at least int he main paper). I looked through some of the proofs and I assume it is the Polyak-Lojasiewicz condition but it should be clarified.


[1] Feldman et al. What Neural Networks Memorize and Why: Discovering the Long Tail via Influence Estimation

[2] Basu et al. Influence Functions in Deep Learning Are Fragile

[3] Bae et al. If Influence Functions are the Answer, Then What is the Question?

**Questions:**

How can one be sure if we can detect memorization with a new, proposed score if we don't train from scratch without that example?

On the theory front, what is the novel technical contribution?

---

> ### Author Response · Authors · 2024-11-23
> **Response Part 1**
>
> We thank the reviewer for their feedback. We address the weaknesses and questions below:
>
> 1. My main concern with the paper lies with the empirical evaluation strategy. The authors validate their proposed metrics (CSL/CSG) against Feldman's [1] subsampled influence scores rather than true leave-one-out training. This is problematic because recent work by Basu et al[2] and Bae et al.[3] has shown that influence functions do not correlate well with actual leave-one-out training for large neural networks. This approximation gets worse for larger networks. Without validation against true leave-one-out training, we cannot verify if their metric actually captures memorization or merely correlates with Feldman's influence scores. Essentially, it would show correlation between two approximations rather than validation against ground truth.
>
>     **A:** We would like to clarify that our paper is **not** about influence functions. Our framework is an entirely separate approach to influence functions, and the two are orthogonal. We outline how our framework differs from influence functions, and as such, the results of Basu et al[2] and Bae et al.[3] do not apply.
>
>     a.  Influence functions aim to estimate the effect of training points on a model’s predictions. This is given by [5]
>
>     $I(z, z_t) = -\nabla_\theta \ell(h(z_t, \hat{\theta}))^\top H_{\hat{\theta}}^{-1} \nabla_\theta \ell(h(z, \hat{\theta}))$.
>
>     This involves:
>         i. The Hessian with respect to model parameters,
>         ii. The inverse of the Hessian, and
>         iii. The gradient loss w.r.t model parameters, typically at the end of training.
>
>     Additionally, influence functions rely on a convex loss approximation.
>
>     b. Our framework, on the other hand, uses the gradient and loss in the **input space** (not the Hessian inverse in the weight space). Additionally, we not only leverage the gradient and loss at the end of training but also throughout the training process, i.e., capturing the **dynamics of learning**. Our method makes no assumptions about the convexity of the loss.
>
>     In summary, we differ because we operate in the input space, incorporate the dynamics of learning, and make more general assumptions than influence functions. This leads to a fundamentally different theoretical framework, and therefore, the results of influence functions from Basu et al[2] and Bae et al.[3] do not apply to our case.
>
>     > "The authors validate their proposed metrics (CSL/CSG) against Feldman's [1] subsampled influence scores rather than true leave-one-out training."
>
>     Since our Theorem suggests that CSL and Feldman's [1] memorization (not influence) are related, we test the tightness of our bound using the experiments.
>
> 2. The theoretical framework assumes that SGD is trained to convergence which isn't really done in most practical settings. On the theoretical side, on its own, this is fine to prove some theorems (with the caveat that they are idealized). But related to the above point, Bae et al. have shown that exactly this property (among others) causes influence functions to not correlate with leave one out retraining. So there is a gap here between the theoretical properties being proven and the practical aspects. In my opinion, such a gap is okay if there is a novel technical contribution on the theoretical side but most proofs seem to follow: Start with SGD convergence assumptions, Use telescoping sums to relate changes across iterations, Apply basic inequalities (Lipschitz, bounded gradients etc.), Arrive at bounds that relate different quantities. Maybe I'm missing something?
>
>     **A:** Regarding influence function please see our response to 1. Regarding the Theoretical Contribution:
>
>     a. We provide a framework to analyze convergence in the **input space**.
>     b. We formally define what constitutes the **sample learning condition**.
>     c. We establish a connection between learning dynamics and **FZ memorization** through bounds.
>     d. Proofs resembling **SGD convergence** are, in fact, focused on the input space. This is non-trivial and depends on (a).
>     e. We would also like bring attention to **Theorems 4.4 – 4.5**, which are follow a different framework and analysis proofs.
>
>     Additionally, our contributions are in identifying the links between various measures (CSL, CSG memorization and learning time) used in the paper, and the strong empirical results that support our theory.

---

> ### Author Response · Authors · 2024-11-23
> **Response Part 2**
>
> 3. On the theoretical side, there is an $\alpha$-adjacency assumption used in the proofs:
>
>     3.1. Requires existence of data points within $\alpha$-balls of each other
>
>     3.2. Authors claim this holds as $\alpha$ is unrestricted, but this makes the assumption essentially meaningless - if can be arbitrarily large, what does the bound tell us? Ideally, I would have like some empirical validation that real datasets satisfy this property for meaningful values as well.
>
>     **A:** Please note the $\alpha$-ball assumption is only used by Theorem 4.5. The other 5 theorems in the paper do not make this assumption. The $\alpha$-ball is an assumption about how well-represented a sample is within the dataset—whether it lies in the mode or in the tail. A small $\alpha$-ball implies the sample is in the mode, while a large one suggests it is in the tail. This concept was previously introduced by Ravikumar et al. [4], and we merely adopt it in our work.
>
>     In summary, our theoretical results show that well-represented samples (in the mode) lead to tighter bounds, while less-represented samples (in the tail) result in looser bounds. This pattern is also evident in Figures 8, 9, and 10, where we plot error bars comparing our proxies with Feldman's[1] memorization results. Feldman [1] demonstrated that less-represented (tail) examples are more likely to be memorized. The error bars grow as memorization increases, consistent with our theory, which is less certain about less-represented samples.
>
>
> 4. The authors use arbitrary thresholds (0.25 for memorization, 0.15 for influence) across all datasets. I did not see a justification for why these specific values were chosen or an analysis of how results change with different thresholds.
>
>     **A:** We are unsure what the reviewer is referring to, as there is no such threshold mentioned or used in the paper.
>
> 5.	No error bars on key results, also. only a single architecture (resNet-18) is used in all experiments.
>
>     **A:** Please note that we have included architecture results in the appendix, which were omitted from the main paper due to space constraints. Please refer to Table 4 in the original submission. Additionally we have added figures (Figures 8, 9, 10) with the error bars in the revised version.
>
> 6.	$\mu$-PL condition is not defined anywhere (at least int he main paper). I looked through some of the proofs and I assume it is the Polyak-Lojasiewicz condition but it should be clarified.
>
>     **A:** We have clarified this in the revised version where we have added it to the "ADDITIONAL BACKGROUND" section. Thank you for bringing this to our attention.
>
> Questions:
>
> How can one be sure if we can detect memorization with a new, proposed score if we don't train from scratch without that example?
>
> **A:** Our theory shows that CSL, is bounded by memorization (FZ memorization). If the bounds are tight we would observe a linear correlation, which is seen in Figures 3 and 4. This validates our theoretical result.  Please note our theory only makes a probabilistic guarantee, with a confidence of 1 − δ in our theorems.
>
> On the theory front, what is the novel technical contribution?
>
> **A:** Please see our response to question 2.
>
> [1] Feldman et al. What Neural Networks Memorize and Why: Discovering the Long Tail via Influence Estimation
>
> [2] Basu et al. Influence Functions in Deep Learning Are Fragile
>
> [3] Bae et al. If Influence Functions are the Answer, Then What is the Question?
>
> [4] Ravikumar et al. "Unveiling Privacy, Memorization, and Input Curvature Links.", ICML 2024
>
> [5] Koh et al. "Understanding black-box predictions via influence functions." In International conference on machine learning, pp. 1885-1894. PMLR, 2017.

---

### Official Review · Reviewer_24L2 · 2024-11-09

**Soundness:** 3
**Presentation:** 3
**Contribution:** 3
**Rating:** 8
**Confidence:** 3

**Summary:**

This paper explores the memorization phenomenon in deep learning by presenting a theoretical framework that links learning time, memorization, and three orders of loss: sample loss, sample gradient, and sample curvature. The authors introduce two proxies, Cumulative Sample Loss (CSL) and Cumulative Sample Gradient (CSG), which exhibit high cosine similarity with established memorization method yet are significantly less computationally expensive. Experiments on deep vision models validate their theory and demonstrate that these proxies achieve state-of-the-art performance in identifying mislabeled or duplicate samples.

**Strengths:**

Originality & Significance:
1. This paper presents a novel theoretical framework that establishes relationships among learning time, stability-based memorization, and memorization proxies: cumulative sample loss, cumulative sample gradient, and input loss curvature. Specifically, the authors demonstrate that (1) learning time exhibits a linear relationship with the three orders of loss, (2) stability-based memorization also follows this linear relationship with these metrics, and (3) learning time and memorization are linearly related.
2. The proposed proxies for memorization, Cumulative Sample Loss (CSL) and Cumulative Sample Gradient (CSG), are significantly more computationally efficient than existing methods. Specifically, the CSL proxy is 4 orders of magnitude less computationally expensive than stability-based memorization (Feldman \& Zhang, 2020) and approximately $14\times$ less costly than input loss curvature (Garg et al., 2024).

Quality:
The theoretical framework is well-supported by detailed assumptions and rigorous proofs. Experiments across multiple deep vision models and datasets validate both the theory and the effectiveness of the proposed proxies for memorization.

Clarity:
Overall, the paper is well-written and well-organized. The "Takeaways" and "Interpreting Theory" sections are particularly effective, as they help clarify the theoretical contributions and experimental results.

**Weaknesses:**

1. Although the theoretical framework is rigorous, it relies on several assumptions, particularly the bounded loss function and the $\lambda$-proximal sample condition, which may not always hold in practice.

2. The experiments are primarily focused on deep vision models, limiting the scope of validation. Testing on other model types or domains, such as natural language processing, would enhance the generalizability of the findings.

**Questions:**

1. The definition of learning time in the main theorems is unclear. It is implicitly mentioned in the sample learning condition (Equation 4), which suggests that learning time depends on a threshold $\tau$ (although in proofs, $\tau$ is chosen as an upper bound in terms of $T$). A more explicit definition or explanation would enhance understanding.
2. In the definition of $\lambda$-proximal, does $T_{ref}$ refer to the learning time of the reference sample $\vec{z}_{\text{ref}}$?

---

> ### Author Response · Authors · 2024-11-23
>
> We thank the reviewer for their feedback. We address the weaknesses and questions below:
>
> 1. Although the theoretical framework is rigorous, it relies on several assumptions, particularly the bounded loss function and the $\lambda$-proximal sample condition, which may not always hold in practice.
>
>     **A:** We would like to clarify that the $\lambda$-proximal is used for mathematical compactness in notation. This has no implications in practice. Since $\lambda$ is unconstrained there will always exist a $\lambda$-proximal sample for some $\lambda$.
>
> 2. The experiments are primarily focused on deep vision models, limiting the scope of validation. Testing on other model types or domains, such as natural language processing, would enhance the generalizability of the findings.
>
>     **A:** We agree with the reviewer, in this paper we focused on vision models. Applying this to language models would improve the generality of the findings, however adapting the results to auto-regressive models is non-trivial and can be explored in the future.
>
> Questions:
>
> 1. The definition of learning time in the main theorems is unclear. It is implicitly mentioned in the sample learning condition (Equation 4), which suggests that learning time depends on a threshold $\tau$ (although in proofs, is chosen as an upper bound in terms of $T$). A more explicit definition or explanation would enhance understanding.
>
>     **A:** We have added a more formal statement to define the learning condition in the revised version. Additionally note that the upper bound for the learning condition can be made independent of $T$, by telescoping the first term in Eq. 23 and using the relation $\ell(w_t) > \ell(w^*)$, thus we get:  $\sum_{t=0}^{T-1} \lVert \nabla_X \ell(\vec{w}_t) \rVert _2^2 \leq \frac{\kappa_t}{\eta} \mathbb{E}_t \left[ \ell(\vec{w}_0) - \ell(\vec{w}^*) \right] + \frac{ \eta  \mathcal{L}\Gamma^2 \Sigma_T}{2}$ which the the upper bound independent of $T$. However, this bound is less tight than the one which depends on $T$. The $T$ dependent bound is more conducive to other proofs.
>
> 2. In the definition of $\lambda$-proximal, does refer to the learning time of the reference sample $\vec{z}_{ref}$?
>
>     **A:** We understand that the definition of $\lambda$-proximal was unclear. We have clarified this in the revised version and restate it below for convenience:
>
>     > There exists a $\lambda$-Proximal iteration $T_{p}$ if $\ell(\vec{w}_{T_{p}}) = (1-\lambda) \ell(\vec{w}_0)$ for some $\lambda$.

---

### Author Response · Authors · 2024-11-23

We sincerely thank all the reviewers for their valuable feedback. We have revised the paper and would like to address a few key points:

1. While our paper focuses primarily on theoretical aspects, we would like to highlight our empirical results, where we achieve state-of-the-art performance in mislabelled and duplicate detection while being significantly more compute-efficient compared to other methods.

2. Based on the reviewers' feedback, we have improved the clarity and interpretability of our theoretical results. Additionally, we would like to emphasize that, although we employ tools from optimization, our work specifically addresses the memorization aspect of deep learning. We present a novel application and new results using these tools. However, our paper is not about optimization. Further, our framework is a totally independent approach and is unrelated to influence functions.

---

### Meta-Review · Area_Chair_cTsS · 2024-12-17

**Metareview:**

This paper proposes an analysis of memorization from a learning dynamics perspective. Unfortunately, the current version has significant issues. Specifically, the theory relies on extremely strong assumptions (e.g., simultaneously bounded, Lipschitz, PL, and stable loss---at least for some of the results), which limits its applicability. This is not even clear in the case of linear parameterizations (e.g., standard OLS is not covered). The applicability needs to be made clear. Further, the results are presented poorly. For instance, in the first result of the paper, Thm 4.1, the "expected learning time for a reference sample" is not defined in or before the result (only in the appendix). Same is true for the "loss estimation variance". Moreover, the theorem suffers from additional clarity issues, as it should be specified that the "confidence" is over the random sampling of the dataset (presumably? not clear given how it is written). Finally, the proofs are very poorly written (e.g., Lemma D.1 starts with a random equation with no explanation). The reviewers identified several additional issues that the authors need to fix. While the paper has merits, it is not ready for publication in the current form.

**Additional Comments On Reviewer Discussion:**

While reviewers acknowledged the clarity of the problem and the novelty of introducing CSL and CSG, they raised concerns about the restrictive assumptions underpinning the theoretical results, such as bounded loss and stability, and the limited validation across domains. Additionally, issues with dense notation, unclear definitions, and weak empirical justification of theoretical claims were highlighted.

---

### Decision · Program_Chairs · 2025-01-22

Reject